# The biogenesis of CLEL peptides involves several processing events in consecutive compartments of the secretory pathway

Nils Stührwohldt[1], Stefan Scholl[2], Lisa Lang[1], Julia Katzenberger[1], Karin Schumacher[2], Andreas Schaller[1]*

[1]Department of Plant Physiology and Biochemistry, Institute of Biology, University of Hohenheim, Stuttgart, Germany; [2]Department of Cell Biology, Centre for Organismal Studies, Heidelberg University, Heidelberg, Germany

**Abstract** Post-translationally modified peptides are involved in many aspects of plant growth and development. The maturation of these peptides from their larger precursors is still poorly understood. We show here that the biogenesis of CLEL6 and CLEL9 peptides in *Arabidopsis thaliana* requires a series of processing events in consecutive compartments of the secretory pathway. Following cleavage of the signal peptide upon entry into the endoplasmic reticulum (ER), the peptide precursors are processed in the cis-Golgi by the subtilase SBT6.1. SBT6.1-mediated cleavage within the variable domain allows for continued passage of the partially processed precursors through the secretory pathway, and for subsequent post-translational modifications including tyrosine sulfation and proline hydroxylation within, and proteolytic maturation after exit from the Golgi. Activation by subtilases including SBT3.8 in post-Golgi compartments depends on the N-terminal aspartate of the mature peptides. Our work highlights the complexity of post-translational precursor maturation allowing for stringent control of peptide biogenesis.

*For correspondence:
Andreas.Schaller@uni-hohenheim.de

Competing interests: The authors declare that no competing interests exist.

## Introduction

Complementing the activity of the classical phytohormones, peptide hormones and growth factors are now recognized as an important class of signaling molecules for long-range signaling and for cell-to-cell communication over short distances, respectively (*Oh et al., 2018*; *Stührwohldt and Schaller, 2019*). In *Arabidopsis thaliana* (hereafter 'Arabidopsis'), there are more than 1000 genes potentially encoding signaling peptides, apparently involved in all aspects of plant growth and development (*Lease and Walker, 2006*; *Ghorbani et al., 2015*; *Tavormina et al., 2015*). There has been remarkable progress in recent years with respect to the characterization of peptide perception and signal transduction mechanisms (*Song et al., 2017*; *He et al., 2018*). The biogenesis of these signaling molecules, on the other hand, is still poorly understood. This is particularly true for the large group of signaling peptides that depend on a series of post-translational modifications (PTMs) for maturation and activation (*Matsubayashi, 2014*; *Stührwohldt and Schaller, 2019*).

Proteolytic processing is required for all post-translationally modified signaling peptides to release the peptide entity from its precursor. Additional PTMs may include tyrosine sulfation, proline hydroxylation, and arabinosylation of the hydroxyproline residue (*Matsubayashi, 2014*; *Stührwohldt and Schaller, 2019*). Tyrosine sulfation is performed by a single tyrosylprotein sulfo-transferase (TPST) that is membrane-anchored in the cis-Golgi (*Komori et al., 2009*). TPST requires aspartate on the amino side of tyrosin for substrate recognition (*Komori et al., 2009*). Tyrosine sulfation is a critical maturation step, as sulfated peptides usually depend on this modification for full activity (*Stührwohldt and Schaller, 2019*). Proline hydroxylation is catalyzed by membrane-anchored prolyl-4-hydroxylases (P4Hs) localized in ER and Golgi compartments. There are 13 P4Hs in

Arabidopsis, some of which were shown to be required for the hydroxylation of extensin and possibly other hydroxyprolin (Hyp)-rich glycoproteins of the cell wall (*Velasquez et al., 2015*). Which of the P4Hs act on signaling peptides, and whether or not they differ in preference for proline in a certain sequence context is still unclear. Proline hydroxylation is a prerequisite for subsequent glycosylation. As the first in a series of glycosylation steps, *L*-arabinose is transferred to the 4-hydroxyl by Golgi-resident Hydroxyproline *O*-arabinosyltransferase (HPAT). HPAT is encoded by three genes in Arabidopsis that are at least partially functionally redundant (*Ogawa-Ohnishi et al., 2013*; *MacAlister et al., 2016*). To what extent differences in substrate specificity of HPATs may contribute to the selection of certain Hyp residues for glycosylation remains to be seen.

The identification of precursor processing proteases lags behind the other PTM enzymes. It was and is still hampered by the large number of possible candidates (907 peptidases are listed in the MEROPS database (release 12.0) for Arabidopsis *Rawlings et al., 2016*), by their generally low expression levels, by functional redundancy, and by the lack of a conserved processing site. As compared to tyrosine sulfation, proline hydroxylation and Hyp arabinosylation which occur at Asp-Tyr, Pro, and Hyp residues, respectively, and in contrast to animal systems, where peptide hormones are typically flanked by pairs of basic residues, there is no consensus motif that would mark the cleavage sites for processing of peptide precursors in plants (*Rawlings et al., 2016*; *Stührwohldt and Schaller, 2019*). We may thus expect that many different proteases with different specificities for cleavage site selection are required for the processing of the many different precursors. Consistent with this notion, precursor processing enzymes have been identified in different classes of proteases, including a metacaspase and a papain-like enzyme among the cysteine peptidases (*Wrzaczek et al., 2015*; *Ziemann et al., 2018*; *Hander et al., 2019*), a carboxypeptidase in the class of the zinc-dependent metallo peptidases (*Casamitjana-Martínez et al., 2003*), and several subtilases (SBTs) among the serine peptidases (*Srivastava et al., 2009*; *Engineer et al., 2014*; *Ghorbani et al., 2016*; *Schardon et al., 2016*; *Stegmann et al., 2017*; *Beloshistov et al., 2018*; *Doll et al., 2020*; *Reichardt et al., 2020*). SBTs thus seem to play a more general role in peptide hormone maturation (*Schaller et al., 2018*).

SBTs constitute a large family of mostly extracellular proteases including e.g. 56 members in Arabidopsis (*Rautengarten et al., 2005*), 86 in tomato (*Reichardt et al., 2018*) and 97 in grapevine (*Figueiredo et al., 2016*). Expansion of the SBT family in plants involved both whole genome and tandem gene duplications with differential neo- and sub-functionalization resulting in many taxon-specific clades (*Taylor and Qiu, 2017*; *Reichardt et al., 2018*). AtSBT6.1 stands out in this diverse family for several reasons. First, unlike most other plant SBTs that are soluble proteins targeted to the cell wall (*Schaller et al., 2018*), AtSBT6.1 is a membrane protein, anchored by a C-terminal membrane-spanning helix to the Golgi and possibly the plasma membrane (*Liu et al., 2007a*; *Ghorbani et al., 2016*). Second, AtSBT6.1 is one of only two Arabidopsis SBTs that originated before the divergence of Metazoa and Viridiplantae, and is functionally conserved between animals and plants (*Taylor and Qiu, 2017*; *Schaller et al., 2018*). Like Site-1-Protease (S1P), its orthologue in humans, AtSBT6.1 (alias AtS1P) cleaves membrane-anchored bZIP transcription factors in the Golgi to facilitate the translocation of their cytoplasmic domain to the nucleus for the induction of ER stress-response genes (*Liu et al., 2007b*; *Liu et al., 2007a*). Furthermore, the cleavage site preference of AtSBT6.1 for the amino acid motives R-(R/K)-X-L or R-(R/K)-L-X (*Srivastava et al., 2009*; *Ghorbani et al., 2016*) is almost identical with the substrate specificity reported for human S1P (R-X-(L/V/I)-X; *Seidah, 2013*).

With its predominant Golgi localization (*Liu et al., 2007a*; *Parsons et al., 2012*), AtSBT6.1 seems predestined to act in concert with TPST, P4Hs and HPATs in the biogenesis of post-translationally modified signaling peptides. Supporting this notion, potential AtSBT6.1/S1P cleavage sites can be found in many peptide precursors including members of the Rapid Alkalinization Factor (RALF), phytosulfokine (PSK), Clavata3/Embryo Surrounding Region (CLE) and CLE-Like (CLEL) families. The latter is also known as Golven (GLV) or Root Meristem Growth Factor (RGF) family and comprises 11 precursor-derived peptides of 13 to 18 amino acids carrying two additional post-translational modifications, i.e. tyrosine sulfation and hydroxylation of the ultimate proline residue (*Matsuzaki et al., 2010*; *Meng et al., 2012*; *Whitford et al., 2012*). We refer to them here as CLEL, because not all family members are involved in root gravitropism (causing the GLV phenotype), or regulating the activity of the root apical meristem (as the name RGF would suggest). However, processing by

AtSBT6.1 has so far only been shown for RALF23 (*Srivastava et al., 2009*; *Stegmann et al., 2017*), and in the particularly interesting case of CLEL6 (GLV1/RGF6) (*Ghorbani et al., 2016*).

AtSBT6.1 was identified as a factor required for CLEL6 function in a screen for *sbt* mutants suppressing the CLEL6-overexpression phenotype (agravitropic root growth and increased hypocotyl elongation) (*Ghorbani et al., 2016*). The protease was shown to cleave the CLEL6 precursor at two canonical AtSBT6.1/S1P cleavage sites (R-R-L-R, R-R-A-L), and both cleavage sites turned out to be relevant for CLEL6 function, the second one even essential. The data indicate that AtSBT6.1 activity is required for the formation of the bioactive CLEL6 peptide (*Ghorbani et al., 2016*). Surprisingly however, AtSBT6.1 cleavage sites are located in the variable part of the CLEL6 and other peptide precursors, considerably upstream of the mature peptide sequence. AtSBT6.1 activity is thus not sufficient and additional unknown protease(s) are required for peptide maturation. Completely unresolved is the question when and where the processing of peptide precursors takes place, particularly in relation to the other PTMs. While the Golgi is an obvious possibility for processing by AtSBT6.1, the enzyme has also been reported at the cell surface (*Ghorbani et al., 2016*) suggesting apoplastic processing of the fully modified precursor as an alternative possibility. This has implicitly been assumed for cell wall-localized SBTs. However, as secretory enzymes they are co-targeted with their potential peptide precursor substrates providing ample opportunity for processing *en route*, in any compartment of the secretory pathway. These are the questions that are addressed here for the CLEL6 and CLEL9 peptide precursors.

## Results

### SBT activity is required for the maturation of CLEL6 and CLEL9 peptides

In order to confirm the involvement of SBTs in the maturation of CLEL6 (GLV1/RGF6), we used the inhibitor-based loss-of-function approach that was previously employed to demonstrate a role for redundant SBTs in the maturation of IDA (Inflorescence Deficient in Abscission) resulting in the shedding of Arabidopsis flower organs after pollination (*Schardon et al., 2016*; *Stührwohldt et al., 2017*; *Stührwohldt et al., 2018*). CLEL9 (GLV2/RGF9) was included in the analysis because it acts redundantly with CLEL6 in the regulation of gravitropic responses (*Whitford et al., 2012*), and because it resembles CLEL6 with respect to predicted processing sites (*Figure 1—figure supplement 1A*). The SBT-specific Extracellular Proteinase Inhibitors (EPIs) 1a and 10 from *Phytophthora infestans* were expressed in transgenic Arabidopsis plants under control of the *CLEL6* or *CLEL9* promoters (*Figure 1—figure supplement 1B*). Inhibition of SBTs by EPIs in tissues where *CLEL6* and *CLEL9* are expressed is expected to phenocopy the *CLEL6/9* loss-of-function phenotype if SBT activity is required for precursor processing and peptide maturation.

Seedlings expressing EPI1a under the control of either the *CLEL6* or the *CLEL9* promoter were impaired in the gravitropic response of the hypocotyl (*Figure 1A,C*). Likewise, hypocotyl gravitropism was inhibited also by the expression of EPI10 controlled by either one of the two *CLEL* promoters (*Figure 1—figure supplement 2*). The same phenotype had been observed when *CLEL6* or *9* were silenced in transgenic plants by artificial micro RNAs (*Whitford et al., 2012*), suggesting that SBT activity is required for CLEL6 and 9 function. However, in contrast to *CLEL6* or *9*-silenced plants (*Whitford et al., 2012*), the gravitropic response of roots was not affected in our EPI-expressing transgenics (*Figure 1—figure supplement 3*). This observation is consistent with the fact that the *CLEL6* and *9* promoters are active in the hypocotyl where they drive the expression of EPI inhibitors in epidermis and cortex (*Whitford et al., 2012*), but not in any part of the primary root (*Fernandez et al., 2013*).

When plants expressing the EPI1a inhibitor were supplied with synthetic CLEL6 or CLEL9 peptides, gravitropism of the hypocotyl was restored to wild-type level (*Figure 1B,D*). Likewise, we observed an impaired gravitropic response in mutants defective in tyrosylprotein sulfotransferase (TPST), and the defect of the *tpst-1* mutant also was alleviated by application of the sulfated CLEL6 or CLEL9 peptides (*Figure 1E,F*). The data indicate that the peptides act downstream of SBT (and TPST) activity, consistent with a role for SBTs (and TPST) in peptide maturation (*Figure 1B,D–F*). The data are fully consistent with findings of *Ghorbani et al. (2016)*, who reported that the activity of

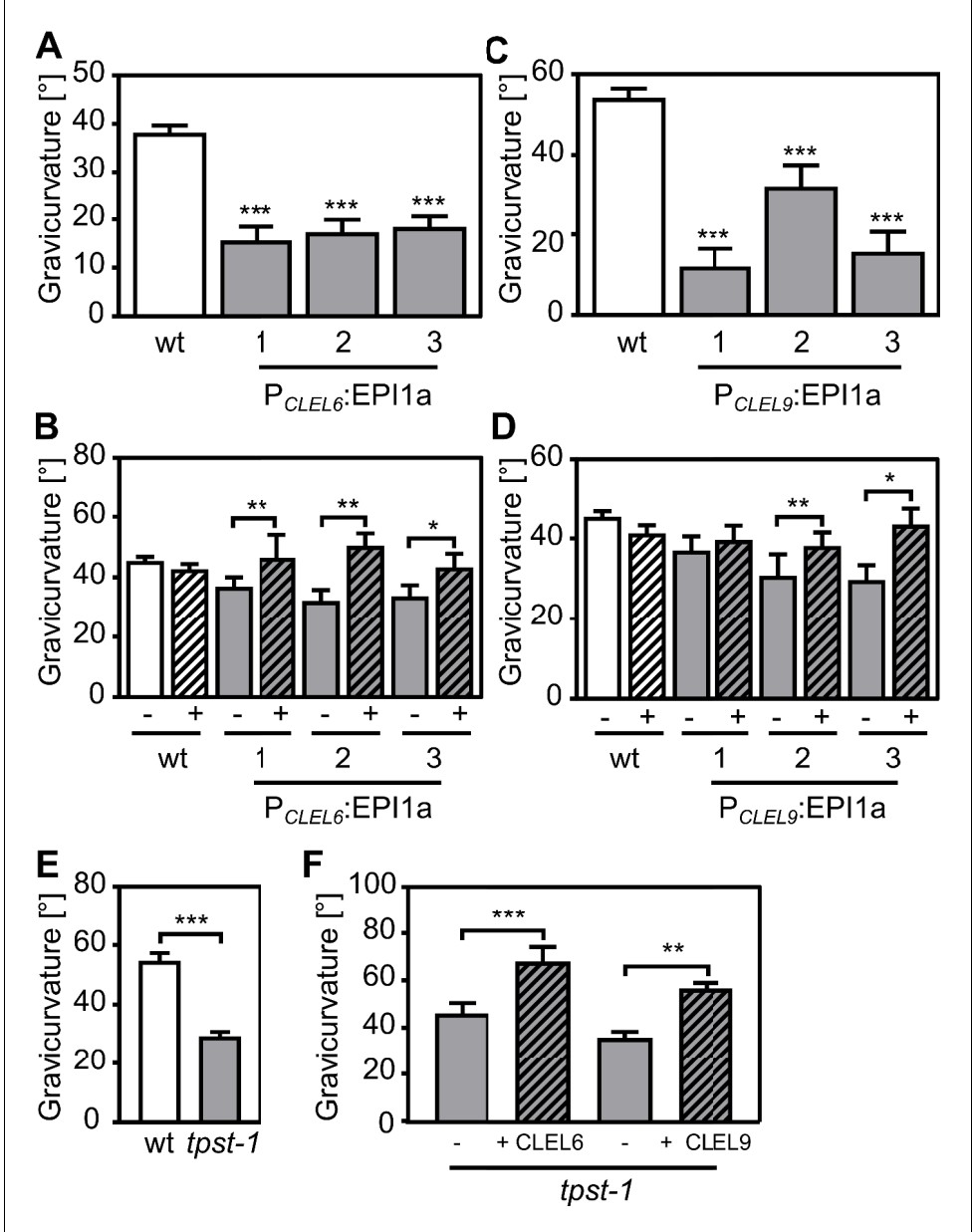

**Figure 1.** Reduced gravicurvature of P_{CLEL6}:EPI1a, P_{CLEL9}:EPI1a and *tpst-1* seedlings is rescued by addition of mature CLEL6 or CLEL9 peptides. Gravicurvature of (**A**) three independent P_{CLEL6}:EPI1a and (**C**) P_{CLEL9}:EPI1a lines (gray bars; promoter constructs are shown in *Figure 1—figure supplement 1*) is significantly reduced in comparison to the wild type (white bars). The same effect was observed when the EPI10 inhibitor was expressed under control of the *CLEL6* or *CLEL9* promoter (*Figure 1—figure supplement 2*). Gravicurvature of P_{CLEL6}:EPI1a and P_{CLEL9}:EPI1a lines is restored to wild-type levels by application of (**B**) 10 nM CLEL6 or (**D**) 300 nM CLEL9 (hatched bars). (**E**) *tpst-1* gravicurvature in comparison to the wild-type control. (**F**) Gravicurvature of *tpst-1* seedlings treated (hatched bars) with CLEL6 (10 nM) or CLEL9 (300 nM) as compared to the untreated *tpst-1* control (gray bars). Seedlings were grown for five days in the dark on ½ MS medium with peptides added as indicated. Plates were rotated 90° and gravicurvature was assessed after two days as the angle of the hypocotyl with the horizontal. Panel A was modified from *Stührwohldt et al. (2017)*. Data are shown for one representative of at least two independent experiments as the mean ± SE (n ≥ 15). *, **, and *** indicate significant differences at p<0.05, p<0.01, and p<0.001, respectively (two-tailed t test). The gravitropic response of roots was not affected in transgenic plants expressing the EPI inhibitors under control of *CLEL* promoters (*Figure 1—figure supplement 3*).

The online version of this article includes the following source data and figure supplement(s) for figure 1:

**Source data 1.** Source data for hypocotyl gravitropic responses shown in *Figure 1* and *Figure 1—figure supplement 2*.
**Figure supplement 1.** Sequence alignment of CLEL6 and CLEL9 and schematic representation of the promoter-EPI constructs.
**Figure supplement 2.** The gravitropic response of the hypocotyl is impaired in P_{CLEL6}:EPI10 and P_{CLEL9}:EPI10 transgenic lines.
**Figure supplement 3.** P_{CLEL6}:EPI1a and P_{CLEL9}:EPI1a lines are not affected in root gravitropism.

SBT6.1 is required for CLEL6 function, and they further indicate that SBTs are required also for the activation of CLEL9.

The CLEL6 precursor comprises two potential S1P (SBT6.1) cleavage sites, RRLR and RRAL (*Figure 2A*), and the second site is necessary for CLEL6 function (*Ghorbani et al., 2016*). However, cleavage by SBT6.1 is not sufficient for CLEL6 formation, since both sites are located considerably upstream of the mature peptide sequence. We thus refer to the cleavage by SBT6.1 as a necessary pre-processing step that precedes peptide activation (*Stührwohldt and Schaller, 2019*). Additional protease(s) are needed to mark the N-terminus and release the fully processed CLEL6 peptide. Whether the final processing for peptide activation also is mediated by SBTs, is still unclear at this time. Also unclear are the subcellular sites of pre-processing and peptide activation, and the sequence of post-translational modification events. These questions were addressed in the following.

## Pre-processing by SBT6.1 in an early Golgi compartment is required for secretion

SBT6.1 is known to be active in the Golgi, where it is required for the processing and activation of membrane-anchored transcription factors and of PMEs (*Liu et al., 2007a*; *Liu et al., 2007b*; *Wolf et al., 2009*; *Sénéchal et al., 2014*), and it was reported also in the cell wall, where it was detected in complex with the Serpin1 inhibitor (*Ghorbani et al., 2016*). To address the question whether pre-processing by SBT6.1 occurs within the secretory pathway or extracellularly, we transiently expressed the CLEL6 precursor fused to sfGFP in *N. benthamiana*. The sfGFP tag was linked to the N-terminus of the precursor, just downstream of the signal peptide (construct named 'Sec' in *Figure 2B*). GFP fluorescence was detected only in the apoplast (*Figure 2D*). On an anti-GFP immunoblot a single band was detected corresponding in size to GFP with part of the precursor up to the first SBT6.1 cleavage site (*Figure 2E*, red asterisk). Processing at this site is thus efficient when the precursor is allowed to pass through the secretory pathway.

When the precursor was equipped with a C-terminal KDEL-motif for ER retention (construct named 'KDEL' in *Figure 2B*), processing was incomplete (*Figure 2E*). In addition to the apoplast, the fluorescence signal was now detected also in the ER and Golgi, as indicated by co-expression of ER (Vma12-mRFP) or Golgi (ManI-mCherry and ST-mCherry for early and late Golgi, respectively) markers (*Figure 2G*). The signal in the ER and early Golgi compartments results from the unprocessed precursor, while the presence of extracellular GFP indicates that some of the precursor was processed to separate GFP from the ER retention signal. Partial processing also is apparent on the western blot, where two additional bands were observed (*Figure 2E*), which we interpret as the precursor processed at the second SBT6.1 site (green asterisk), and the unprocessed precursor (black asterisk), respectively.

ER retention by the KDEL-motif is mediated by Golgi-resident K/HDEL-receptors, which effect retrograde transport of soluble ER proteins from the Golgi back to the ER (*Pelham, 1988*; *Phillipson et al., 2001*; *Silva-Alvim et al., 2018*). Cleavage by SBT6.1 may thus occur either in the ER or in the Golgi. However, processing was abolished when anterograde ER-to-Golgi vesicle transport was inhibited by addition of brefeldin A (*Nebenführ et al., 2002*; *Figure 2C*). These observations indicate that exit from the ER is required for cleavage by SBT6.1, and we conclude that SBT6.1 acts in the Golgi, likely in an early Golgi compartment. This conclusion was confirmed by fusing the CLEL6 precursor to the N-terminal membrane anchor of ß−1,2-xylosyltransferase (XylT), which is sufficient to target reporter proteins to the *medial* Golgi (*Pagny et al., 2003*; *Figure 2H*). The same three cleavage products were observed as for the KDEL-tagged precursor at somewhat different ratios (*Figure 2E*) indicating that cleavage by SBT6.1 occurs before the precursor reaches the *trans* Golgi network (TGN).

To assess the relevance of cleavage by SBT6.1 for processing and secretion, we masked both cleavage sites (*Figure 2A*; RRLR and RRRAL) by alanine substitutions as described by *Ghorbani et al. (2016)* and analyzed the effect on the processing pattern of the transiently expressed precursor (compare constructs 'Sec' and 'Sec^m' in *Figure 2F*). The central band corresponding to the second cleavage site was lost for the Ala-substituted precursor confirming that SBT6.1 is responsible and necessary for this cleavage event. This may not be the case for the first cleavage event, as the corresponding band was still observed for the Ala-substituted precursor, suggesting that another protease may jump in when cleavage by SBT6.1 is prevented, or a different

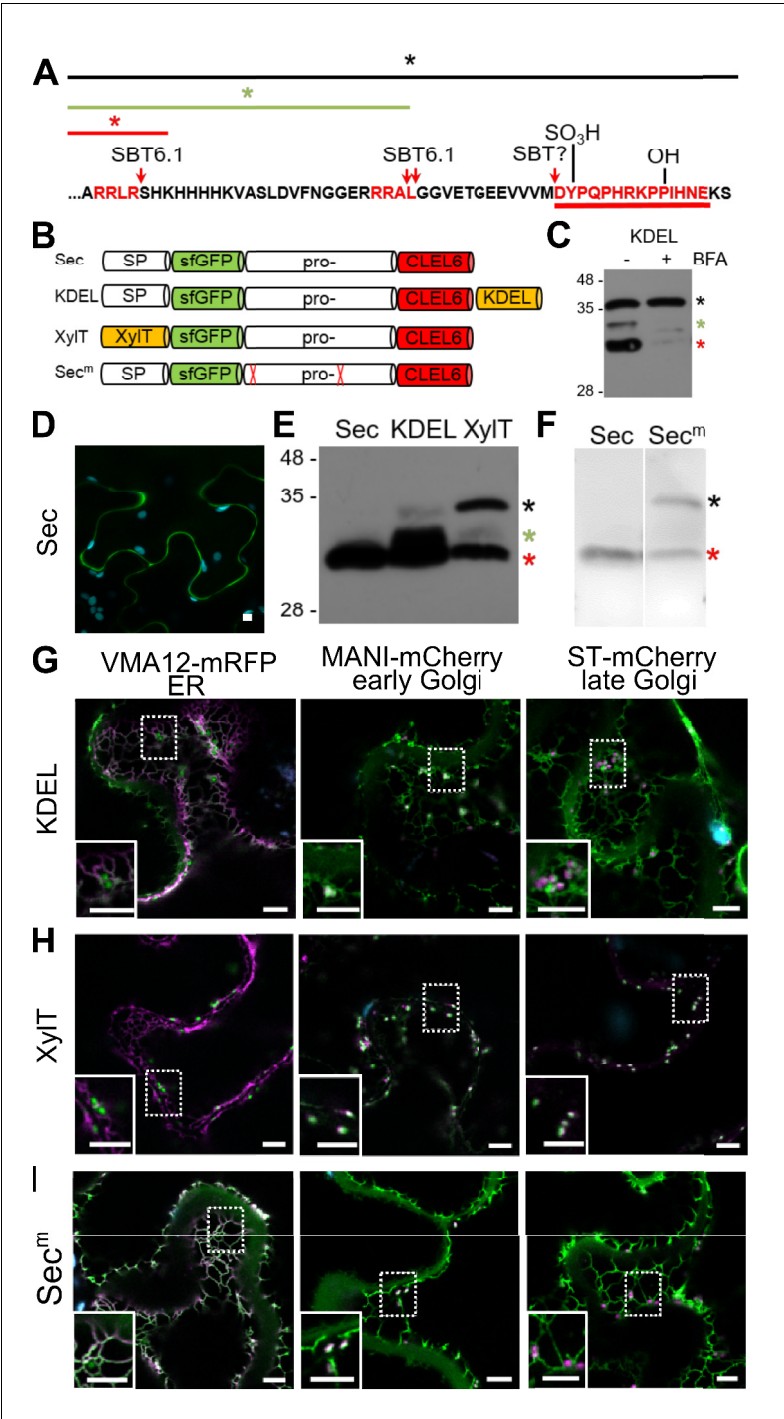

**Figure 2.** Pre-processing of CLEL6 by SBT6.1 in the *cis*-Golgi is required for peptide maturation and secretion. (A) Sequence of the CLEL6 precursor without the signal peptide. Proteolytic processing sites and other post-translational modifications are indicated, mature CLEL6 peptide sequence underlined. Colored lines and asterisks represent the size of the processed forms of CLEL6 observed in panels C, E, and F. (B) Schematic representation of expression constructs used for subcellular localization studies. (C) Processing of the ER-restrained CLEL6 (KDEL) construct with (+) or without (-) BFA treatment, analyzed by anti-GFP immunoblotting. Different processed forms of the precursor are marked by colored asterisks as shown in panel A. (E, F) Immunoblot analysis of ER-restrained (KDEL) and Golgi-localized (XylT) precursor processing compared to the secreted form (Sec) and the precursor lacking the two SBT6.1 cleavage sites (Sec^m). Fully and partially processed precursors are indicated by the colored asterisks as defined in panel A. (D, G–I) Co-localization of the different fusion proteins with ER (VMA12-mRFP) and Golgi (ManI-mCherry and ST-mCherry) markers analyzed by fluorescence microscopy. Pictures show an overlay of the green (500–550 nm) and red (610–670 nm) fluorescence channels. The dotted areas are shown in higher magnification in the insets. Scale bars represent 5 μm.

protease cleaves the mutant cleavage site. Interestingly, comparing *Figure 2D and I*, we observed that secretion of the Ala-substituted precursor is reduced compared to the wild-type. For the Ala-substituted precursor, the GFP fluorescence signal was observed in both ER and Golgi in addition to the apoplastic space suggesting that cleavage by SBT6.1 may facilitate continued passage along the secretory pathway and, hence, additional post-translational modifications in post-Golgi compartments. This observation may explain why pre-processing by SBT6.1 is required for CLEL6 function in vivo (*Ghorbani et al., 2016*), despite the fact that this cleavage event does not produce the mature peptide.

## The cleavage for final activation occurs in a post-Golgi compartment by aspartate-dependent subtilase SBT3.8

After pre-processing of CLEL6 by SBT6.1 in the Golgi, additional processing at the N-terminus is required for maturation and activation. To localize this processing event subcellularly, we used an N-terminally sfGFP-tagged deletion construct of the CLEL6 precursor (Δ-Sec) lacking both SBT6.1 processing sites (*Figure 3A*). Again, we analyzed a secreted version (Δ-Sec), one that was equipped with a C-terminal KDEL-motif for ER retention (Δ-KDEL) and one that was anchored to the Golgi membrane (Δ-XylT, *Figure 3B*). On an anti-GFP immunoblot a single band was detected for Δ-Sec corresponding in size to the precursor processed at the N-terminal maturation site (*Figure 3C*, blue asterisk). Interestingly, for Δ-Sec some of the GFP signal was observed in the cell wall, in addition to the ER and a weak signal in the late Golgi (ST-mCherry marker in *Figure 3D*). Secretion of this construct, like that of Sec$^m$ (*Figure 2I*), is thus reduced compared to wild-type Sec (*Figure 2D*), suggesting that the propeptide, in addition to propeptide cleavage by SBT6.1 (*Figure 2I*), may contribute to efficient passage through the secretory pathway.

In contrast to Δ-Sec, the unprocessed precursors were observed for both Δ-KDEL and Δ-XylT as single larger bands on the immunoblot (*Figure 3C*, black asterisk). The GFP signal for the C-terminally KDEL-tagged deletion (Δ) construct was found exclusively in the ER (*Figure 3E*). Retention of the N-terminal GFP tag in the ER confirmed that processing did not occur, indicating that the maturation step is located further downstream in the secretory pathway. For Δ-XylT the apparently unprocessed precursor (*Figure 3C*) co-localized exclusively with Golgi markers (*Figure 3F*), suggesting a post-Golgi compartment or, at the latest, the apoplastic space as the site for CLEL6 maturation. Maturation late in the secretory pathway was also observed for CLEL9. Similar to CLEL6, the Δ-KDEL and Δ-XylT constructs for CLEL9 were not processed and were retained in the ER and Golgi, respectively (*Figure 3—figure supplement 1*). For the Δ-Sec construct that is allowed to proceed beyond the Golgi, on the other hand, the smaller, processed product was generated (*Figure 3—figure supplement 1*). Our data thus indicate that both CLEL6 and CLEL9 mature after exit from the Golgi, in the TGN, in secretory vesicles, or in the apoplastic space.

All CLEL peptide precursors including both CLEL6 and 9 share a conserved aspartate upstream of the sulfated tyrosine (*Figure 1—figure supplement 1A*). To test whether this aspartate is necessary for peptide processing and/or activity, we generated site-directed D-to-A mutants of both CLEL6 (D71A) and CLEL9 (D66A) and compared processing to the corresponding wild-type versions (*Figure 4A,B*). As compared to the fully processed Δ-Sec constructs of CLEL6 and CLEL9, the larger unprocessed form was observed for the Δ-Sec D71A and Δ-Sec D66A mutants, indicating that the aspartate is indeed required for processing (*Figure 4A,B*). When fused to the XylT Golgi anchor, the processing-resistant D71A and D66A mutants exhibited the same apparent molecular weight as the wild-type Δ-XylT constructs (*Figure 4A,B*), thus confirming that the band produced from the wild-type Δ-XylT construct of CLEL6 corresponds to the full-length precursor, despite its faster migration as compared to the unprocessed Δ-KDEL band (*Figure 3C*).

A bioassay was then used to assess whether the aspartate and aspartate-dependent processing are required for the formation of bioactive CLEL peptides in planta. The full-length Sec CLEL6 and CLEL9 constructs were transiently expressed in *N. benthamiana* and any peptides produced from these precursors were extracted in apoplastic washes. The activity of these peptides was tested in the *tpst-1* mutant, which is devoid of endogenous sulfated peptides (*Figure 4C*). When *tpst-1* seedlings were treated with cell wall extracts of plants expressing sfGFP fusions of wild-type CLEL6 or CLEL9 precursors (GFP fluorescence was determined as a measure of protein expression, and equal amounts of GFP were used), the gravitropic response was restored to wild-type levels indicating the formation of bioactive CLEL6 and CLEL9 peptides (*Figure 4C*). In contrast, there was no bioactivity

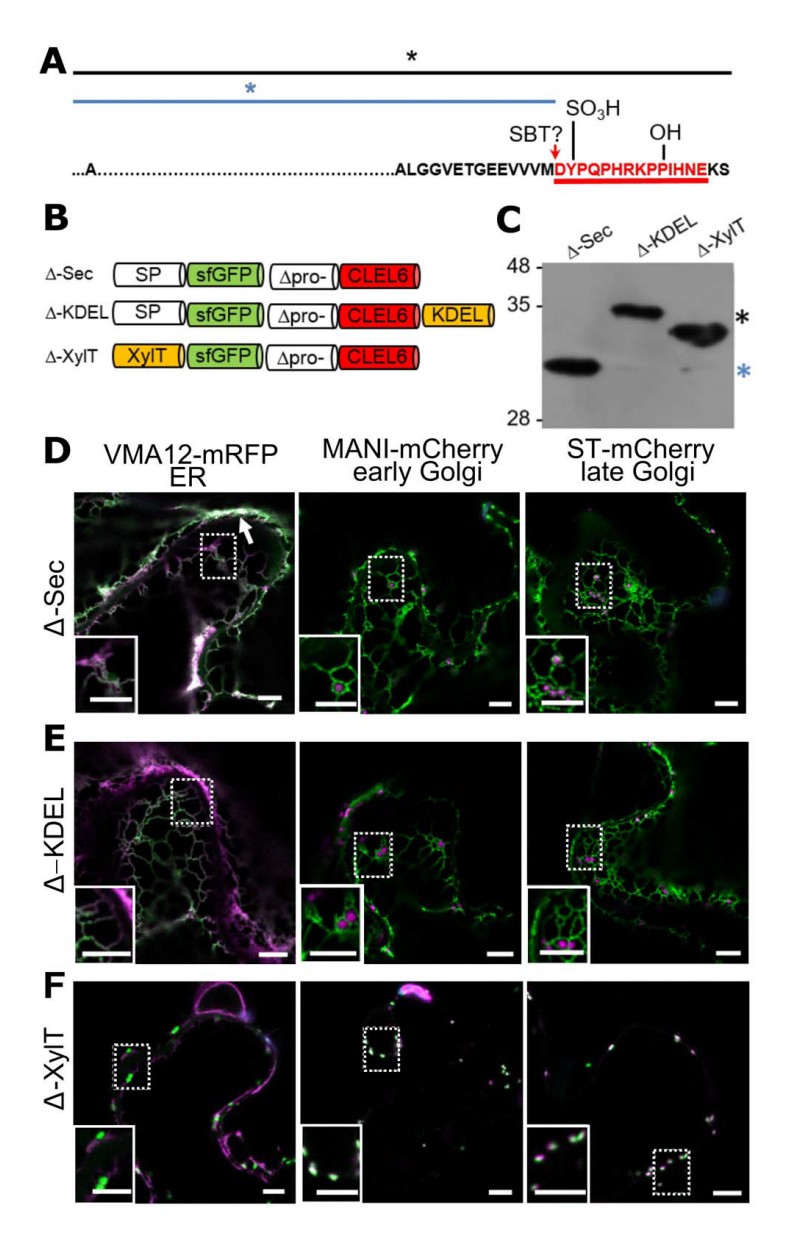

**Figure 3.** N-terminal maturation of CLEL6 occurs in a post-Golgi compartment (the subcellular localization of CLEL9 maturation is analyzed in *Figure 3—figure supplement 1*). (**A**) Sequence and post-translational modification sites of Δ-CLEL6 constructs lacking the prodomain region encompassed by the two SBT6.1 cleavage sites; mature CLEL6 peptide sequence underlined. Black and blue lines and asterisks were included to represent the unprocessed and processed forms of the precursor, respectively. (**B**) Schematic representation of expression constructs used to localize the subcellular compartment of CLEL6 maturation. (**C**) Processing of the secreted (Δ-Sec), ER-restrained (Δ-KDEL) and Golgi-localized (Δ-XylT) constructs analyzed by anti-GFP immunoblotting. Unprocessed and processed forms of the precursor are indicated by the black and blue asterisks, respectively. (**D–F**) Co-localization of the different fusion proteins with ER (VMA12-mRFP) and Golgi (ManI-mCherry and ST-mCherry) markers analyzed by fluorescence microscopy. Pictures show an overlay of the green (500–550 nm) and red (610–670 nm) fluorescence channels. The dotted areas are shown in higher magnification in the insets. The white arrow marks apoplastic localization; scale bars represent 5 μm.

The online version of this article includes the following figure supplement(s) for figure 3:

**Figure supplement 1.** N-terminal maturation of CLEL9 occurs in a post-Golgi compartment.

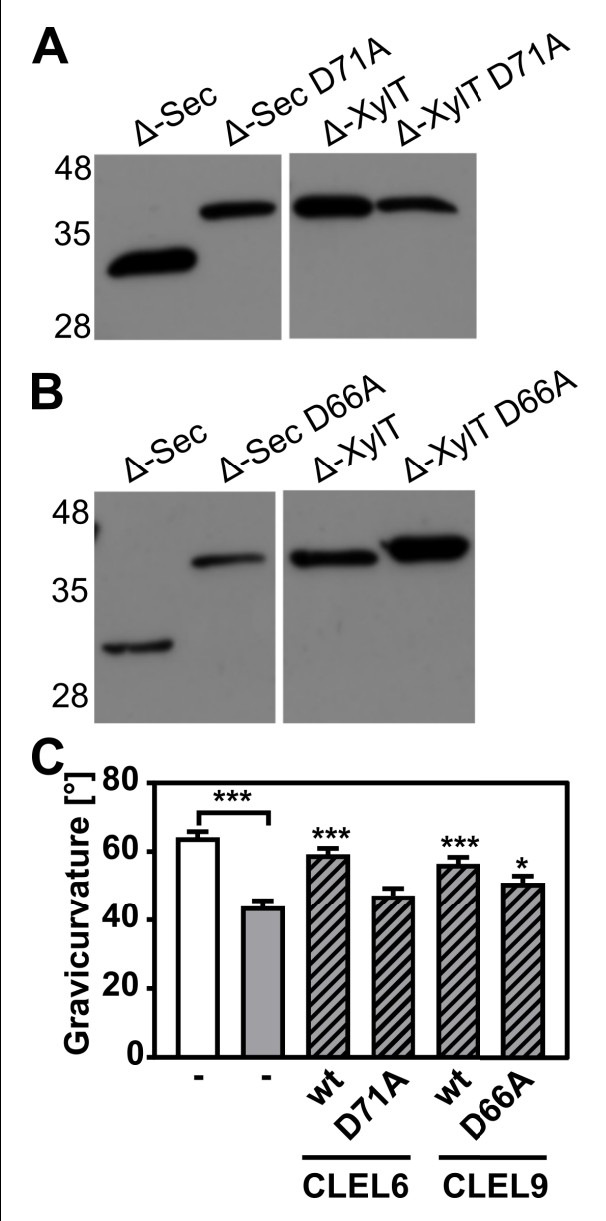

**Figure 4.** N-terminal maturation and the formation of bioactive CLEL6 and 9 peptides are aspartate-dependent. (A, B) The relevance of the N-terminal aspartate for precursor processing was analyzed on anti-GFP immunoblots for the secreted (Δ-Sec) and Golgi-anchored (Δ-XylT) constructs by alanine substitution (D71A and D66A for the CLEL6 and CLEL9 precursors, respectively). Aspartate-dependent processing requires SBT activity (**Figure 4— figure supplement 1**). (C) Complementation of the gravitropic response of the *tpst-1* mutant by CLEL peptides produced in planta. The gravitropic response of the *tpst-1* mutant (gray bar) was restored to wild-type levels (white bar) by treatment with cell wall extracts from plants expressing the CLEL6 or CLEL9 precursors (hatched bars). Activity was much reduced for extracts from plants expressing the D71A and D66A precursor mutants. Seedlings were grown for five days in the dark on ½ MS medium with peptides added as indicated. Plates were rotated 90° and gravicurvature was assessed after two days as the angle of the hypocotyl with the horizontal. Data show the average of three independent experiments as the mean ± SE (n ≥ 103). Unless otherwise indicated * and *** indicate significant differences to the *tpst-1* control (gray bar) at p<0.05 and p<0.001, respectively (two-tailed t test).

The online version of this article includes the following source data and figure supplement(s) for figure 4:

**Source data 1.** Source data for hypocotyl gravitropic responses shown in **Figure 4**.

**Figure supplement 1.** SBT activity is required for N-terminal maturation of CLEL6 and CLEL9.

in cell wall extracts of plants expressing the alanine-substituted CLEL6 precursor, and reduced bioactivity in extracts from plants expressing the D66A-CLEL9 precursor (*Figure 4C*). The data confirm the importance of the aspartate residue for peptide maturation.

The protease(s) required in addition to SBT6.1 for the final maturation step and activation of CLEL6 and CLEL9 was hitherto unknown. To test a potential involvement of SBTs, as suggested by the impaired gravitropic response of the hypocotyl in seedlings expressing EPI1a or EPI10 under the control of either the *CLEL6* or the *CLEL9* promoter (*Figure 1A,C*; *Figure 1—figure supplement 2*), the flag-tagged EPI10 inhibitor was co-expressed with the Δ-Sec CLEL6 and CLEL9 constructs in *N. benthamiana* (*Figure 4—figure supplement 1*). As compared to the single, fully processed band that was observed again for Δ-Sec CLEL6 and Δ-Sec CLEL9, co-expression of EPI10 reduced the efficiency of processing resulting in additional bands corresponding to the unprocessed CLEL6 and CLEL9 precursors (*Figure 4—figure supplement 1*). The data indicate that on top of SBT6.1, another SBT activity is required, directly or indirectly, for the maturation of CLEL peptides.

To identify candidate SBTs, we reasoned that proteases required for the maturation of CLEL peptides might be up-regulated in the *tpst-1* mutant, as a compensatory response to the deficiency in bioactive sulfated peptides. We thus analyzed the expression of the 56 Arabidopsis *SBT* genes in *tpst-1* hypocotyls in comparison to the wild type and found that four (SBTs 1.7, 3.7, 3.8, and 4.14) were up-regulated in the *tpst-1* background (*Figure 5A*). Interestingly, SBT3.8 (At4g10540) has recently been described as an aspartate-dependent protease, and selectivity for aspartate at the cleavage site was reported to be pH dependent (*Chichkova et al., 2018*). Therefore, to test a possible involvement of SBT3.8 in the maturation of CLEL peptides, we expressed the enzyme with a C-terminal His tag and purified it from tobacco cell wall extracts (*Figure 5B*). The activity of recombinant SBT3.8 was analyzed in comparison to a mock-purification from control plants using a synthetic, N-terminally extended CLEL6 peptide (eCLEL6) as substrate. eCLEL6 included seven precursor-derived amino acids in addition to the mature CLEL6 sequence. eCLEL6 was processed in a SBT3.8-dependent manner to produce the mature DYPQPHRKPPIHN peptide at pH 5.5 (*Figure 5B*, *Figure 5—figure supplement 1*). In contrast, at pH 7.0, this cleavage was not observed and mature CLEL6 was not produced (data not shown). pH-dependent cleavage of eCLEL6 at acidic pH suggests the *trans*-Golgi or the cell wall as possible compartments for final processing of the CLEL6 precursor. Consistent with this proposition, we localized sfGFP-tagged SBT3.8 to the apoplastic space, both in agro-infiltrated tobacco leaves and in stably transformed Arabidopsis lines (*Figure 5C*).

To test whether processing by SBT3.8 depends on aspartate at the cleavage site, the CLEL6 precursor lacking the N-terminal signal peptide and a corresponding D71A mutant were expressed in *E. coli* and digested with recombinant SBT3.8 in vitro. Consistent with the presence of two aspartate residues in the CLEL6 precursor (*Figure 2A*), two cleavage products were observed (marked by asterisks in *Figure 5D*), corresponding to cleavage after D71 (upper band) and D47 (lower band). The upper band was not observed for the D71A mutant, confirming that cleavage at this site is aspartate-dependent (*Figure 5E*). To further test whether processing by SBT3.8 at D71 is affected by sulfation of the neighboring tyrosine (Y72), we produced recombinant tyrosine-sulfated CLEL6 (sulfoCLEL6) in *E. coli* using an expanded genetic code (*Liu et al., 2009*). Briefly, a CLEL6-(His)6 expression construct with an amber stop replacing the Y72 codon was co-expressed in *E. coli* with a suppressor tRNA recognizing the amber stop, and a matching aminoacyl-tRNA synthetase specific for sulfo-tyrosine (*Liu et al., 2009*). Co-translational incorporation of sulfo-tyrosine that was chemically synthesized and added to the growth medium resulted in the Y72-sulfated CLEL6-(His)6 precursor. If sulfo-tyrosine is not incorporated, translation terminates at the amber stop resulting in a truncated precursor lacking the C-terminal His-tag. The His-tag thus allowed for selective purification of the sulfated precursor, that was then tested as a substrate of SBT3.8. The tyrosine-sulfated CLEL6 precursor was cleaved by SBT3.8 in a time-dependent manner similar to the non-sulfated precursor, indicating that tyrosine sulfation does not affect SBT3.8-mediated aspartate-dependent processing (*Figure 5—figure supplement 2A*).

Finally, we compared CLEL6 precursor processing in wild-type plants and in *sbt3.8* loss-of-function mutants. The recombinant CLEL6 precursor was cleaved efficiently by exudates prepared from wild-type plants, but not by *sbt3.8* exudates (*Figure 5—figure supplement 2B*). These data support a role for SBT3.8 in CLEL6 maturation in vivo. However, we did not observe any defect in the gravitropic response of the hypocotyl in *sbt3.8* mutants, suggesting that there are other proteases acting

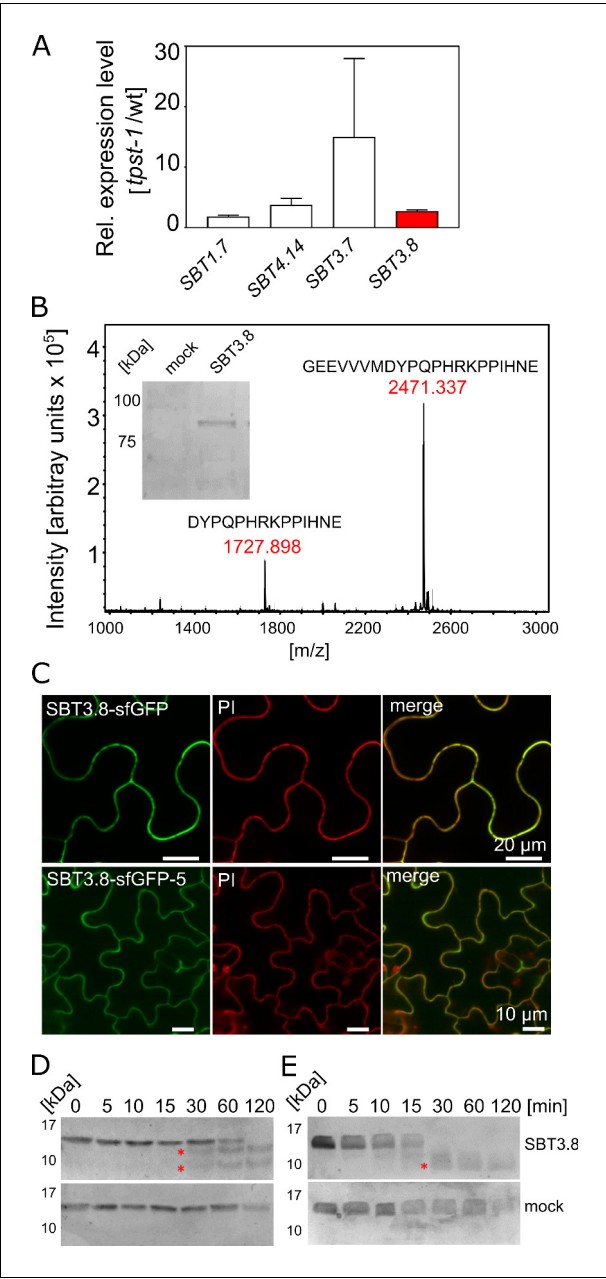

**Figure 5.** SBT3.8 is localized extracellularly and requires Asp at the cleavage site for peptide activation. (**A**) qPCR analysis of *SBT* genes that are expressed at higher levels in etiolated *tpst-1* hypocotyls in comparison to wild type. Relative SBT mRNA levels were determined after normalization to three reference genes (*Actin2, EF* and *Tubulin*). (**B**) MALDI-TOF analysis of eCLEL6 (GEEVVVMDYPQPHRKPPIHNE) cleavage by SBT3.8. Masses of eCLEL6 and mature CLEL6 (DYPQPHRKPPIHNE) are indicated. SDS-PAGE analysis of purified SBT3.8 and the mock control is shown in the insert. The control digest is shown in *Figure 5—figure supplement 1*. (**C**) Subcellular localization of SBT3.8-sfGFP transiently expressed in tobacco leaves (upper panels) and in stably transformed Arabidopsis plants (lower panels) under the control of the CaMV 35S promoter. Cell walls were stained with propidium iodide vacuum-infiltrated five min prior to imaging. (**D**) Cleavage of the CLEL6 precursor by SBT3.8. The recombinant CLEL6 precursor was incubated with purified SBT3.8 (top) or the mock control (bottom) for the time indicated. Cleavage was detected by Coomassie staining after SDS-PAGE. Red asterisks mark SBT3.8 cleavage products (**E**) Cleavage of the D71A CLEL6 precursor mutant by SBT3.8. The analysis was performed as described for (**D**). Note missing cleavage product in in (**E**) compared to (**D**).

The online version of this article includes the following source data and figure supplement(s) for figure 5:

**Figure supplement 1.** Control digest of eCLEL6.

*Figure 5 continued on next page*

*Figure 5 continued*

**Figure supplement 2.** Cleavage of the CLEL6 precursor by SBT3.8 is not affected by tyrosine sulfation and impaired in the *sbt3.8* mutant.

**Figure supplement 2—source data 1.** Source data for hypocotyl gravitropic responses shown in *Figure 5—figure supplement 2*.

redundantly with SBT3.8 in aspartate-dependent CLEL6 maturation (*Figure 5—figure supplement 2C*; *Figure 6*).

## Discussion

Using an inhibitor-based approach targeting SBT function at the level of enzyme activity rather than gene expression, we confirmed that SBTs are required for the gravitropic response of etiolated Arabidopsis seedlings (*Figure 1A,C*; *Figure 1—figure supplement 2*). The loss-of-function phenotype of EPI1a and EPI10-expressing transgenic plants was complemented by application of the mature CLEL6 and CLEL9 peptides (*Figure 1B,D*), indicating that SBT activity is required upstream of the peptides, consistent with a role in peptide formation. SBT6.1 was previously shown to be necessary for CLEL6 function in a screen for *sbt* mutants suppressing the overexpression phenotype of the CLEL6 precursor (*Ghorbani et al., 2016*). Consistently, two SBT6.1 cleavage sites were identified in the precursor, and the second site was found indispensable for CLEL6 activity (*Ghorbani et al., 2016*).

Addressing the sequence and subcellular sites of maturation events we show here that several consecutive processing steps are required for the biogenesis of CLEL peptides in Arabidopsis. Cleavage of CLEL precursors by SBT6.1 constitutes the first obligatory processing step in peptide maturation (not considering the co-translational cleavage of the signal peptide). However, the sub-cellular site of SBT6.1-mediated processing remained unresolved. We show here that SBT6.1 cleaves CLEL precursors soon after exit from the ER in an early Golgi compartment (*Figure 2*), indicating that the reported plasma-membrane localization (*Ghorbani et al., 2016*) is irrelevant for the maturation of CLEL6 and 9.

Site-directed mutagenesis of SBT6.1 cleavage sites impaired secretion of the CLEL precursors, as some of the signal got stuck in the ER and the Golgi (*Figure 2I,F*). The variable pro-region of the precursor including the SBT6.1 cleavage sites is thus important for secretion. Similarly in the animal field, neurotrophins are synthesized as larger pro-proteins that need proteolytic processing to yield mature and biologically active neurotrophic factors, which play important roles in the development, maintenance and regeneration of the nervous system (*Suter et al., 1991*). For brain-derived and glial cell-line derived neurotrophic factors (BDNF and GDNF, respectively) the cleavable prodomain was found to be required for post-Golgi trafficking. Sorting of BDNF and GDNF to secretory granules depends on sorting receptors of the Vps10p (vacuolar protein-sorting 10 protein) family, sortilin and sorLA, respectively (*Chen et al., 2005*; *Geng et al., 2011*). Sortilin also facilitates prodomain-dependent export of hydrophobic conotoxins from the ER, by allowing them to escape ER quality control mechanisms (*Conticello et al., 2003*). Likewise, the prodomain of CLEL peptides may interact with unidentified sorting receptors to facilitate secretion or, alternatively, cleavage of the prodomain by SBT6.1 may provide a point of quality control, before the now partially processed precursor is allowed to leave the Golgi for final activation.

The requirement of SBT6.1-mediated processing for secretion provides an explanation for the perplexing finding that SBT6.1 cleavage sites and, by inference, cleavage by SBT6.1 are required for CLEL6 activity (*Ghorbani et al., 2016*), despite the fact that cleavage at these sites does not produce the active peptide. Additional processing is obviously required. We show here that this second obligatory cleavage event marking the N-termini of the mature CLEL6 and 9 peptides takes place in a post-Golgi compartment, i.e. in the TGN, in secretory vesicles, or ultimately in the apoplastic space. Our data suggest that cleavage by SBT6.1 allows for continued passage of partially processed (pre-activated) precursors through the secretory pathway, and thereby facilitates subsequent post-translational modifications in the Golgi (sulfation, proline hydroxylation) and post-Golgi compartments (proteolytic maturation) (*Figure 6*).

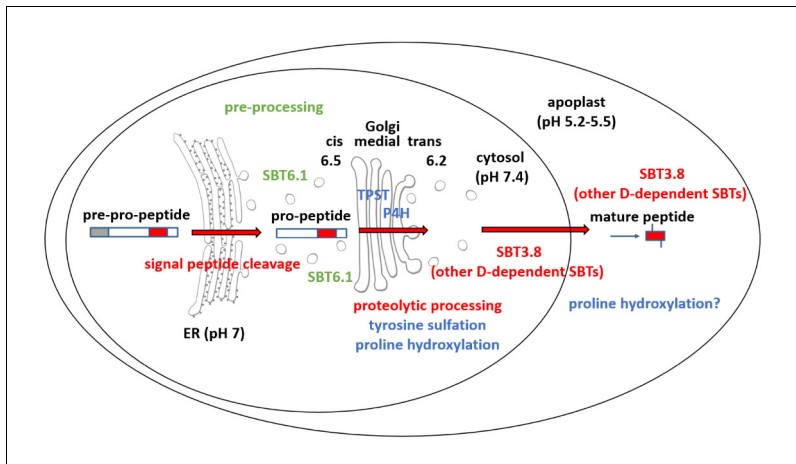

**Figure 6.** Maturation of the CLEL6 precursor in the secretory pathway. As a first processing step, the signal peptide is cleaved off upon entry into the ER. After exit from the ER, the precursor is processed by SBT6.1 in the cis-Golgi at two S1P cleavage sites within its variable prodomain. Still in the Golgi, the peptide moiety is tyrosine-sulfated by TPST, and proline hydroxylated. Candidate proline-4-hydroxylases have been identified, but it is still unclear which of these enzymes is responsible for peptide modification. There is also circumstantial evidence for proline-4-hydroxylase activity in the cell wall (*Stührwohldt et al., 2018*). N-terminal maturation by Asp-dependent SBT3.8 occurs late in the secretory pathway in a post-Golgi compartment, or in the apoplastic space. The figure was modified and updated from *Stührwohldt and Schaller (2019)*.

The second obligatory processing event and formation of the bioactive peptide hinged on the aspartate residue at the cleavage site (D71 for CLEL6 and D66 for CLEL9; *Figure 4A–C*). This processing step was inhibited by the SBT-specific inhibitor EPI10 (*Tian et al., 2005*; *Figure 4—figure supplement 1*), indicating that cleavage at the N-terminus of CLEL6 and 9 peptides also is mediated by SBT(s), particularly by Asp-dependent SBT(s). Here, we identified SBT3.8 as one of the enzymes that ultimately activate CLEL6 and CLEL9. Unlike SBT6.1 and 6.2, all other members of the plant SBT family are secretory enzymes (*Schaller et al., 2018*) that co-migrate with potential pro-peptide substrates through the secretory pathway towards the cell wall as their final destination, thus providing ample opportunity for interaction *en route*. Nonetheless, cleavage does not occur before the partially processed precursor leaves the Golgi (*Figure 3*). We see two possible explanations for this apparent discrepancy. First, the precursor may not be fit for cleavage, or second, the protease may not be active before exit from the Golgi.

The first scenario implies that post-translational modifications in earlier compartments are mandatory for subsequent SBT-mediated cleavage. Interestingly, tyrosin sulfation by TPST is known to depend on an adjacent aspartate residue (*Hanai et al., 2000*; *Komori et al., 2009*). Therefore, if the apparent aspartate-dependency of the N-terminal maturation step is only indirect, and the SBT responsible for this processing event rather needs sulfo-tyrosin for cleavage site recognition, post-translational modification by Golgi-resident TPST would be a prerequisite for SBT-mediated cleavage. However, this scenario is not supported by our data, since the CLEL6 precursor produced in *E. coli* was efficiently processed by SBT3.8, despite the lack of tyrosine sulfation. In fact, the sulfated und non-sulfated CLEL6 precursors were found to be cleaved similarly by SBT3.8, indicating that tyrosin-sulfation does not affect cleavage site recognition.

Alternatively, it may be the control of SBT activity rather than co-localization in the secretory pathway that determines the subcellular site of proteolytic pro-peptide activation. In general, SBT activity is controlled by the prodomain that acts as an intramolecular chaperone for folding, and as an inhibitor of the mature enzyme (*Meyer et al., 2016*). SBT zymogens remain inactive until the prodomain is cleaved off auto-catalytically and subsequently released (*Anderson et al., 2002*; *Cedzich et al., 2009*; *Janzik et al., 2000*; *Meyer et al., 2016*). Prodomain-mediated inhibition and latency of the zymogen are pH-dependent, and broken in a compartment-specific manner as the pH drops along the secretory pathway. In the well-studied case of furin, for example, the prodomain is released in a second autocatalytic cleavage event, which does not occur before the zymogen

reaches the acidic environment of the TGN (*Anderson et al., 2002*). Similarly, SBT3 from tomato also requires the acidic pH of post-Golgi compartments for prodomain cleavage and activation (*Meyer et al., 2016*). These findings may explain why the second obligatory cleavage event by SBT3.8 and, possibly, additional redundant activities does not occur before peptide precursors exit the Golgi.

As an alternative to the pH-dependent release of the inhibitory prodomain, protease activity and cleavage site selectivity may be controlled by pH directly. Indeed, we did not observe cleavage of the N-terminally extended eCLEL6 peptide by SBT3.8 at neural pH, whereas mature CLEL6 was produced at pH 5.5 (*Figure 5B*). We conclude that processing of the CLEL6 precursor by SBT3.8 requires an acidic environment which is encountered only after exit from the Golgi. The extracellular localization of SBT3.8 (*Figure 5C*) is fully consistent with precursor processing and release of mature CLEL peptides in post Golgi compartments including the apoplastic space.

## Materials and methods

### Key resources table

| Reagent type (species) or resource | Designation | Source or reference | Identifiers | Additional information |
|---|---|---|---|---|
| Gene (*Arabidopsis thaliana*) | *SBT3.8* | The Arabidopsis Information Resource (TAIR) | At4g10540 | amplified from genomic DNA |
| Strain, strain background (*Escherichia coli*) | BL21-RIL | Agilent | | |
| Strain, strain background (*Escherichia coli*) | BL21 pEVOL SYRS | *Liu et al., 2009* | | |
| Genetic reagent (*Arabidopsis thaliana*) | sbt3.8 | Nottingham Arabidopsis Stock Center | SALK_052039 | |
| Genetic reagent (*Arabidopsis thaliana*) | tpst-1 | Nottingham Arabidopsis Stock Center | SALK_009847 | |
| Genetic reagent (*Arabidopsis thaliana*) | $P_{CLEL6}$:EPI1a; $P_{CLEL9}$:EPI1a | this paper | | Transgenic lines expressing EPI1a under control of the *CLEL6* or CLEL9 promoter |
| Genetic reagent (*Arabidopsis thaliana*) | $P_{CLEL6}$:EPI10; $P_{CLEL9}$:EPI10 | this paper | | Transgenic lines expressing EPI10 under control of the *CLEL6* or *CLEL9* promoter |
| Genetic reagent (*Agrobacterium tumefaciens*) | C58C1; GV3101 | Community resource | NCBI:txid176299 | GV3101 with and without pSOUP helper plasmid |
| Antibody | anti-GFP, polyclonal, rabbit | Thermo Fisher Scientific | Cat# A-11122 | 1:10000 |
| Antibody | anti-His, monoclonal, mouse | Dianova | Cat# DIA-900–200 | 1:10000 |
| Antibody | anti-FLAG-HRP, monoclonal, mouse | Sigma-Aldrich | Cat# A8592 | 1:5000 |
| Antibody | Goat-anti-rabbit IgG, HRP conjugate | Calbiochem | Cat# 401315 | 1:10000 |
| Antibody | Goat-anti-mouse IgG, HRP conjugate | Calbiochem | Cat# 402335 | 1:10000 |

*Continued on next page*

*Continued*

| Reagent type (species) or resource | Designation | Source or reference | Identifiers | Additional information |
|---|---|---|---|---|
| Recombinant DNA reagent | pCR2.1-Topo | Life Technologies | Cat# K4510-20 | PCR cloning vector |
| Recombinant DNA reagent | pART7, pART27 | *Gleave, 1992* | | plant transformation vectors |
| Recombinant DNA reagent | sfGFP | *Pédelacq et al., 2006* | | used as template for cloning |
| Recombinant DNA reagent | pETDuet1 | Novagen/Merck | Cat# 71146 | for expression of recombinant proteins |
| Recombinant DNA reagent | pMS119EH-sfGFP | *Pross et al., 2016* | | |
| Recombinant DNA reagent | EPI1a, EPI10 | *Schardon et al., 2016* | | codon-optimized, with plant signal peptide |
| Recombinant DNA reagent | pGreen0229 | *Hellens et al., 2000* | | plant transformation vector |
| Recombinant DNA reagent | VMA12-mRFP | *Viotti et al., 2013* | | ER marker |
| Recombinant DNA reagent | VHP1Pro: ManI-mCherry | this paper | | Golgi marker; see Materials and methods section and *Supplementary file 2* |
| Recombinant DNA reagent | VHP1Pro:ST-mCherry | this paper | | Golgi marker; see Materials and methods section and *Supplementary file 2* |
| Recombinant DNA reagent | Sec | this paper | | sfGFP-tagged CLEL6/9 expression constructs; *Figure 2* |
| Recombinant DNA reagent | KDEL | this paper | | sfGFP-tagged CLEL6/9 expression constructs with ER retention signal; *Figure 2* |
| Recombinant DNA reagent | XylT | this paper | | sfGFP-tagged CLEL6/9 expression constructs with XylT membrane anchor; *Figure 2* |
| Recombinant DNA reagent | Sec$^m$ | this paper | | sfGFP-tagged CLEL6/9 expression constructs, SBT6.1 cleavage sites mutated; *Figure 2* |
| Recombinant DNA reagent | Δ-Sec | this paper | | sfGFP-tagged CLEL6/9 pro-domain deletion constructs; *Figure 3* |
| Recombinant DNA reagent | Δ-KDEL | this paper | | sfGFP-tagged CLEL6/9 prodomain deletion constructs with ER retention signal; *Figure 3* |
| Recombinant DNA reagent | Δ-XylT | this paper | | sfGFP-tagged CLEL6/9 prodomain deletion constructs with XylT membrane anchor; *Figure 3* |
| Recombinant DNA reagent | SBT3.8-sfGFP | this paper | | Expression construct for SBT3.8 C-terminally fused with sfGFP; *Figure 5* |
| Peptide, recombinant protein | SBT3.8 | this paper | | Recombinant, His-tagged SBT3.8, purified from *N. benthamiana* cell wall extracts |

*Continued on next page*

*Continued*

| Reagent type (species) or resource | Designation | Source or reference | Identifiers | Additional information |
|---|---|---|---|---|
| Peptide, recombinant protein | CLEL6 | PepMic | | DsYPQPHRKPPIHNE |
| Peptide, recombinant protein | CLEL9 | PepMic | | DMDsYNSANK KRPIHNR |
| Peptide, recombinant protein | eCLEL6 | PepMic | | GEEVVVMDYPQP HRKPPIHNE |
| Peptide, recombinant protein | proCLEL6; CLEL6 precursor | this paper | | purified from *E. coli* BL21-RIL cells |
| Peptide, recombinant protein | CLEL6-D71A | this paper | | Site-directed mutant of the CLEL6 precursor purified from *E. coli* BL21-RIL cells |
| Peptide, recombinant protein | sulfoCLEL6 | this paper | | purified from *E. coli* BL21-pEVOL SYRS cells |
| Commercial assay or kit | Ni-NTA agarose | Qiagen | Cat# 30210 | |
| Software, algorithm | GraphPad Prism | Graphpad | | preparation of figures and statistical analyses |
| Software, algorithm | ImageJ | ImageJ | | analysis of gravitropic response |

## Plant material and growth conditions

For growth experiments in axenic culture, Arabidopsis seeds were surface-sterilized in 70% ethanol for 15 min, washed in 100% ethanol and laid out in rows on square plates containing 0.5 x MS (Murashige-Skoog) medium, 1% sucrose and 0.38% gelrite. Seeds were stratified for two days at 4°C and grown for 5 days in the dark. For quantitative analysis of gravitropic responses, 5 day old vertically grown seedlings were rotated for 90° in the dark and grown for further two days. The bending angle was measured using ImageJ (http://rsbweb.nih.gov/ij/). All experiments were carried out at least three times with similar results. If indicated, media were supplemented with synthetic CLEL6 (DY(SO₃H)PQPHRKPPIHNE) or CLEL9 (DMDY(SO₃H)NSANKKRPIHNR) peptides (PepMic, Suzhou, China) at the indicated concentrations. The *sbt3.8* loss-of-function mutant has been described before *Rautengarten et al. (2005)*.

## Generation of expression constructs

The PCR primers used for amplification of CLEL6/9 precursors and tags are listed in *Supplementary file 1*. As a general strategy, PCR products with flanking restriction sites were first cloned into pCR2.1-Topo (Life Technologies, Carlsbad, CA) and verified by sequencing (Macrogen, Amsterdam, The Netherlands). Using the flanking restriction sites (*Supplementary file 1*), the inserts were mobilized from pCR2.1-Topo and cloned into pART7 (*Gleave, 1992*) between the cauliflower mosaic virus (CaMV) 35S promoter and terminator. The entire expression cassette was then transferred into the *Not*I site of pART27 (*Gleave, 1992*) for transient expression in plants. Strains C58C1 or GV3101 were used for Agrobacterium-mediated expression. More specifically, for the generation of constructs with sfGFP (*Pédelacq et al., 2006*) inserted between the N-terminal signal peptide or the XylT35 membrane anchor and the CLEL propeptide sequences, overlapping PCR was used to fuse the ORFs of the CLEL6 signal peptide or the first 35 amino acids of 1,2-xylosyltransferase (*Pagny et al., 2003*) to the 5'-end of sfGFP. For the generation of CLEL6 and CLEL9 constructs in

C-terminal fusion to sfGFP, the propeptide ORFs were amplified by PCR and cloned into the *Eco*RI site of pART7. The C-terminal KDEL sequence for ER retention was included in the PCR primers. Orientation was tested by PCR and sequencing. The ORF of sfGFP with C-terminal hexa-His tag was cut out from pMS119EH-sfGFP (*Pross et al., 2016*) with *Bam*HI and *Hin*dIII, and subcloned into the *Bam*HI and *Xba*I sites of pART7, in translational fusion with the propeptide ORFs (*Hin*dIII and *Xba*I sites were blunted). CLEL6 and CLEL9 were N-terminally coupled to sfGFP using *Eco*RI and *Bam*HI. For constructs expressing EPI1a and EPI10 under the control of the *CLEL6* or *CLEL9* promoters, EPI1a and EPI10 constructs described by *Schardon et al. (2016)* were used as a starting point. EPI1a, modified with a flag tag insertion between the signal peptide and the inhibitor, and EPI10-flag ORFs were amplified by PCR to include *Eco*RI and *Xho*I restriction sites and ligated into the corresponding restriction sites of pGreen0229, upstream of the *nptII* terminator sequence. *CLEL6* and *CLEL9* promoters (*Whitford et al., 2012*) were PCR-amplified with terminal *Not*I and *Eco*RI restriction sites, and ligated into the corresponding sites of pGreen0229, upstream of EPI1a and EPI10, respectively. For SBT3.8 with six C-terminal histidines, the SBT3.8 ORF was amplified from genomic DNA with a reverse primer including six His codons and first cloned into pCR2.1-Topo. *Eco*RI sites from pCR2.1 were used for ligation into pART7. Orientation was verified by sequencing. For SBT3.8-sfGFP constructs, the SBT3.8 ORF was amplified from the construct above and ligated into pART7 by *Eco*RI and *Bam*HI in translational fusion with C-terminal sfGFP. Plasmids were transformed into GV3101 containing the pSOUP helper plasmid and transformed into *Arabidopsis thaliana* Col 0 by floral dip (*Clough and Bent, 1998*). Transgenic lines were selected on glufosinate or kanamycin and homozygous lines in the T3 or T4 generation were used in further experiments.

The expression construct for the VMA12-mRFP ER marker has been described previously (*Viotti et al., 2013*). The expression constructs $VHP1_{Pro}$:ManI-mCherry and $VHP1_{Pro}$:ST-mCherry for Golgi markers were generated using the GreenGate cloning system (*Lampropoulos et al., 2013*). GreenGate modules used are listed in *Supplementary file 2*. To generate new entry modules, fragments were PCR amplified from pre-existing plasmids, *Arabidopsis thaliana* Col-0 genomic DNA or cDNA with 'Phusion High-Fidelity DNA-Polymerase' (Thermo Scientific, Waltham, MA). After purification, PCR-products were digested with *Eco*31I-HF (Thermo Scientific) to open module specific overhangs. Fragments were then ligated in *Eco*31I-opened and purified entry vectors. Presence and sequence of inserts were verified via restriction digest and sequencing.

For expression in *E. coli*, the ORF of CLEL6 lacking the predicted signal peptide was amplified by PCR from a previous construct and cloned into the *Nco*I restriction site of pETDuet1 (Novagen/Merck KGaA, Darmstadt, Germany). Correct orientation and translational fusion with the C–terminal His-tag was verified by sequencing. Expression in *E. coli* BL21 was induced by 1 mM IPTG for two hours. His-tagged CLEL6 precursor was purified from bacterial extracts by metal chelate affinity chromatography on Ni-NTA Agarose (Qiagen, Hilden, Germany) according to the manufacturer's recommendations. After ultrafiltration (30 kDa molecular weight cutoff), the recombinant protein in the filtrate was further purified by size exclusion chromatography (NGC chromatography system with Enrich SEC 650 column, BioRad, Munich, Germany) in 50 mM $NaH_2PO_4$ /$Na_2HPO_4$, pH 5.5, 10 mM NaCl. The D71A mutant was created by site-directed mutagenesis, confirmed by sequencing, and purified as above. For co-translational incorporation of sulfo-tyrosin, the Y72 codon of the pro-CLEL6 ORF in pETDuet1 was replaced with an amber stop (UAG) by site-directed mutagenesis and confirmed by sequencing. The sulfoCLEL6 expression construct was transformed into electro-competent BL21 pEVOL SYRS carrying an amber suppressor t-RNA and a cognant sulfo-tyrosine specific aminoacyl-tRNA synthetase (*Liu et al., 2009*). Sulfo-tyrosine was synthesized chemically as described by *Liu et al. (2009)*. sulfoCLEL6 expression in *E. coli* BL21 pEVOL SYRS was induced by 1 mM IPTG overnight, and sulfo-tyrosine was added to the growth medium at 10 mM. His-tagged sulfoCLEL6 was purified as described above.

## Transient expression in *N. benthamiana* and protein extraction

*A. tumefaciens* strains C58C1 and GV3101 were used for transient expression in *N. benthamiana*. Bacteria were grown on plates containing appropriate antibiotics (rifampicin, tetracycline and spectinomycin for C58C1 and gentamycin and spectinomycin for GV3101) at 28°C and were washed off the plates in 10 mM MES, pH 5.6 containing 10 mM $MgCl_2$. A blunt syringe was used to infiltrate the bacterial suspension supplemented with 150 µM acetosyringone into the leaves. For total protein extraction, leaves were harvested two to three days after infiltration into liquid nitrogen and ground

to a fine powder. The powder was thawed in 50 mM Tris/HCl, pH 7.5, 100 mM NaCl and 10 mM β-mercaptoethanol containing 0.5% Triton X-100 and proteinase inhibitor mix P (#39103, SERVA Electrophoresis GmbH, Heidelberg, Germany). The extracts were centrifuged (16.000 g, 4°C, 10 min) and the supernatant was kept at 4°C until usage at the same day, or frozen at −20°C.

## Extraction of apoplastic proteins and purification of SBT3.8

Five days after agro-infiltration, the leaves were harvested and vacuum (70 mbar)-infiltrated with 50 mM $NaH_2PO_4/Na_2HPO_4$, pH 7, 300 mM NaCl. Apoplastic washes were harvested by centrifugation at 1100 x g. In order to obtain SBT3.8 in sufficient amounts and purity for assays, metal chelate affinity chromatography on Ni-NTA Agarose (Qiagen) was performed according to the manufacturer's recommendations. The eluates were dialyzed against 50 mM sodium phosphate buffer pH 7.0 or 5.5, 10 mM NaCl and used for enzyme activity measurements. For the empty-vector control, apoplastic extracts from mock-infiltrated plants were subjected to the same purification scheme. To collect exudates from wild type plants and the *sbt3.8* mutant, seedlings were grown in submerged culture in 0.5 x MS (Murashige-Skoog) medium with 1% sucrose for ten days as described by *Ohyama et al. (2009)*. Under these conditions, seedlings release their extracellular protein content into the medium (*Ohyama et al., 2009*). Apoplastic proteins were enriched by ultrafiltration with a molecular weight cutoff of 5 kDa. Exudates were used at a protein equivalent of 500 ng for cleavage assays.

## CLEL6 digest and MALDI TOF analysis

Extended CLEL6 peptide (eCLEL6; GEEVVVMDYPQPHRKPPIHNE, 2 μM) was digested with recombinant SBT3.8 or the negative control for 2 hr until the reaction was stopped by addition of 1% TFA. Reactions were performed at pH 5.5 and pH 7.0 in 50 mM potassium phosphate buffer, 10 mM NaCl. 1.5 μl of the samples were mixed with an equal volume of the crystallization matrix (5 mg/ml α-cyano-4-hydroxy-trans-cinnamic acid in 50% acetonitrile, 0.1% TFA) on the MALDI target, and mass spectra were recorded with a AutoflexIII mass spectrometer (Bruker Daltonics) in the reflector mode with external calibration (Peptide Calibration Standard II; Bruker Daltonics). Flex Analysis 3.0 was used for data analysis with a mass tolerance of 50 ppm for ions. Recombinant CLEL6 and the D71A mutant were digested with SBT3.8 in 50 mM $NaH_2PO_4/Na_2HPO_4$, pH 5.5, 10 mM NaCl for the time indicated and separated by SDS-PAGE.

## Hypocotyl bioassay with in vivo produced peptides

C-terminally sfGFP-tagged CLEL6/9 expression constructs were infiltrated into tobacco leaves. Leaves were harvested after five days and apoplastic extracts were obtained as above. GFP concentration was determined spectro-fluorimetrically using a Spark microplate reader (Tecan; Crailsheim, Germany; excitation 395 nm, emission 509 nm). Equivalent amounts of cell wall extract (equal amounts of GFP) were directly applied to the growth media of etiolated seedlings, and the gravitropic response was analyzed as described before. For the control, cell wall extracts were prepared from empty-vector infiltrated plants and used at the largest volume of experimental samples.

## Immunodetection

Proteins were separated by SDS-PAGE or Tris Tricine PAGE. For western blots, proteins were transferred to nitrocellulose membranes using standard procedures. Polyclonal anti-GFP antibodies (1:10000; A-11122, Thermo Fisher Scientific, Waltham, Massachusetts, USA), monoclonal anti-His (1:10000; Dianova, Hamburg, Germany) or anti-Flag antibodies (1:5000; Sigma-Aldrich, Taufkirchen, Germany) directly coupled to horseradish peroxidase were used for immunodetection, followed by enhanced chemiluminescence detection with an Odyssey Fc imager (Li-COR Biotechnology, Homburg, Germany).

## Fluorescence microscopy

Agro-infiltrated leaves of *N. benthamiana* were observed with a TCS SP5 II inverted Confocal Laser Scanning Microscope (Leica Microsystems, Wetzlar Germany) using a HCX PL APO lambda blue 63.0 × 1.20 water immersion objective (Leica Microsystems). sfGFP was excited with the 488 nm line of the VIS-Argon laser; for mRFP/mCherry the 561 nm line generated by a VIS-DPSS 561 laser was

used. Emission was detected at 500–550 nm for sfGFP and 610–670 nm for mRFP/mCherry with HyD hybrid detectors (Leica Microsystems) in standard operation-mode. Autofluorescence was detected between 700–800 nm with identical laser settings as used for sfGFP-mRFP/mCherry image recording. Images were adjusted in brightness and processed using 'Mean' Filter with a pixel radius of 0.1 with ImageJ software version 1.51 s (National Institute of Health). Propidium iodide (10 mg/ml; Thermo Fisher Scientific) was vacuum-infiltrated into leaves five min prior to imaging (extinction 515 nm/emission 595 nm).

## qPCR analysis

RNA was isolated from approximately 50 hypocotyls of 5 day-old etiolated seedlings as previously described with minor modifications (*Kutschmar et al., 2009*). cDNA was synthesized from 0.8 µg of total RNA with oligo dT primers and RevertAid Reverse Transcriptase (Thermo Fisher Scientific). SBT primers for qPCR analysis are listed in *Supplementary file 3*. Quantitative PCRs (total volume 25 µl) were performed in biological triplicates with two technical repeats on the obtained cDNAs using a CFX96 Real-Time PCR Detection system (BioRad). Primer efficiencies and optimal primer concentrations were determined experimentally. qPCR was performed with Taq polymerase expressed in and purified from *E. coli* and SYBR-Green (Cambrex Bio Science Rockland Inc; Rockland, ME, USA). Relative SBT mRNA levels were determined after normalization to three reference genes (*Actin2*, *EF* and *Tubulin*) using the optimized ΔCT method by *Pfaffl (2001)*.

## Acknowledgements

The work was supported by the Deutsche Forschungsgemeinschaft [SFB 1101 to Andreas Schaller (project D06) and Karin Schumacher (project A02)]. Authors thank Lhana Stein and Isolde Schuck for excellent technical support. We also thank Rainer Waadt for contributing the pGGA-VHP1 entry module and The Scripps Research Institute for the plasmid pEVOL-SYRS.

## Additional information

### Funding

| Funder | Grant reference number | Author |
| --- | --- | --- |
| Deutsche Forschungsge-meinschaft | SFB1101 Project D06 | Andreas Schaller |
| Deutsche Forschungsge-meinschaft | SFB1101 Project A02 | Karin Schumacher |

The funders had no role in study design, data collection and interpretation, or the decision to submit the work for publication.

### Author contributions

Nils Stührwohldt, Conceptualization, Supervision, Investigation, Visualization, Writing - review and editing; Stefan Scholl, Investigation, Visualization; Lisa Lang, Julia Katzenberger, Investigation; Karin Schumacher, Supervision, Funding acquisition; Andreas Schaller, Conceptualization, Funding acquisition, Writing - original draft, Project administration, Writing - review and editing

### Author ORCIDs

Nils Stührwohldt (iD) https://orcid.org/0000-0003-4166-3786
Karin Schumacher (iD) http://orcid.org/0000-0001-6484-8105
Andreas Schaller (iD) https://orcid.org/0000-0001-6872-9576

### Decision letter and Author response

Decision letter https://doi.org/10.7554/eLife.55580.sa1
Author response https://doi.org/10.7554/eLife.55580.sa2

## Additional files

### Supplementary files

• Supplementary file 1. List of primers for cloning.

• Supplementary file 2. GreenGate modules used including (A) the list of modules, and (B) sources of templates and cloning primers.

• Supplementary file 3. Primers used for qPCR analysis.

• Transparent reporting form

### Data availability

All data generated or analysed during this study are included in the manuscript and supporting files. Source data have been provided for Figure 1 panels A to F and Figure 1—figure supplements 2B and 2C, for Figure 4C, and for Figure 5—figure supplement 2.

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
