## [Decision Letter]

**Acceptance summary:**

In Arabidopsis and other plants, CLEL peptides serve as ligands for major signaling pathways. Bioactive CLEL peptides must be processed from larger precursor proteins. In this work, the authors identify multiple cleavage steps occurring in consecutive compartments of the secretory pathway, and candidate subtilisin-like proteases, SBT6.1 and SBT3.8, required for these steps. This work highlights the complexity of post-translational precursor maturation allowing for stringent control of peptide biogenesis in plants.

**Decision letter after peer review:**

[Editors’ note: the authors submitted for reconsideration following the decision after peer review. What follows is the decision letter after the first round of review.]

Thank you for submitting your work entitled "The biogenesis of CLEL peptides involves several processing events in consecutive compartments of the secretory pathway" for consideration by *eLife*. Your article has been reviewed by three peer reviewers, one of whom is a member of our Board of Reviewing Editors, and the evaluation has been overseen by a Senior Editor. The reviewers have opted to remain anonymous.

Our decision has been reached after consultation between the reviewers. Based on these discussions and the individual reviews below, we regret to inform you that your work will not be considered further for publication in *eLife*.

There was consensus that the context and rationale for the study was well laid out and that there is a need for better mechanistic understanding of how CLEL peptides are processed. The extent to which this study definitively advanced our knowledge beyond that of previous studies, however, was considered by all reviewers to be fairly modest. Certainly identification of the SBT capable of driving the next processing reaction would have made this work much more compelling.

Reviewer #1:

This is an engagingly written article about CLEL processing. The CLE-like family of peptides have been known for about 20 years; initially via genetics in Arabidopsis. There are many biological functions ascribed to this family, particularly during development-shoot meristems, root meristems, embryos all employ family members, and these proteins have also been found and studied in Maize, tomato, etc. The challenge of this family is that the active peptide are only 12 AA long and are processed and heavily modified. What I didn't appreciate until this manuscript was that, although many of the modification enzymes have been identified, the initial proteolytic processing is fairly uncharacterized. For a family as critical as this one, missing the first step is a big deal!

Subtilisin proteases (SBT) are found in all forms of life. In plants, there is a large family of SBTs, but most are extracellular. This manuscript focuses on SBT6.1, one of the few TM-domain forms, and one that is also thought to have emerged before the plant/animal split. SBT6.1 was recently shown to be required for activity of CLEL6 (Ghorbani, 2016). So, this manuscript must add something new. What's new is that they show where the first SBT6.1 processing takes place and, they tracked down a second processing step that need to take place in the Golgi and they found the residue in CLEL that is needed for this second step. They then do an assay that suggests another SBT (or something that can be inhibited by EPI, anyway) is needed later on, but don't know what it is.

The approach relies on several technical innovations, some of which I find very clever and others (though also clever) I am less convinced of the specificity of.

e.g., I like that they set up a transient cleavage assay (N. benthi) with GFP attached to the CLEL precursors + various tags in which they could monitor process through the secretory system with live cell imaging and also by Immunoblot against GFP to see where processing occurred.

I am a little worried about overexpression of a pathogen-derived EPI, in that its specificity isn't completely clear. I do like the genetic experiments of showing that EPIs driven with CLEL6 or CLEL9 promoters leads to a phenotype similar to loss of the CLEL6/9 peptides themselves, and can be suppressed by supplying synthesized, processed CLEL.

Strengths: This is well written and I am now convinced that it's important to understand the processing of the CLEL peptides and that the processing is going to be complicated. The assays are clever and the experiments appear to be done rigorously. The data are presented in a clear and straightforward manner in the figures.

Weaknesses: Relies on EPIs being highly specific (and also, highly effective). Biochemical assays are in a non-native system, although I recognize that it is likely not technically feasible to do the assays in Arabidopsis and they also include bioassays in Arabidopsis, so I am less concerned about this leading to completely artifactual results.

Although they demonstrated that it IS important for the peptides to be processed at the site they identified, is this a surprise based on previous work?

The manuscript does a nice job of demonstrating that a late Golgi SBT is likely to be involved, but then can't supply an answer as to which one it is, and I felt like there was no attempt to use other data sources (co-expression, for example) to narrow down the possibilities. All in all, I found this a bit too preliminary for an e*Life* paper.

The Discussion was quite long and speculative relative to the compact data section. It seems as if the authors are addressing some points of contention in the field, and I (as an outsider) don't come to the story with a particular bias that needs to be overcome. I will defer to others who are closer in the field.

Reviewer #2:

In this manuscript, the authors studied biogenesis mechanisms of Arabidopsis CLEL6 and CLEL9 peptides by expressing their GFP fusion proteins using N. benthamiana transient expression system followed by Western blotting. Cumulative results suggested that SBT6.1-mediated cleavage within the variable domain is a prerequisite for continued passage through the secretory pathway and subsequent proteolytic maturation after exit from the Golgi. Furthermore, the authors proposed that the cleavage for final activation occurs in a post-Golgi compartment by an aspartate dependent subtilases.

Because the maturation mechanisms of plant peptide hormones is still poorly understood, the authors' efforts will contribute to expand our knowledge of the peptide hormone biogenesis. Indeed, their findings that cleavage within the variable domain is a prerequisite for continued passage through the secretory pathway and subsequent proteolytic maturation are interesting. However, some of the conclusions are not supported by the results, especially the assumed involvement of aspartate residue in the cleavage for final activation.

1) The authors concluded that aspartate-dependent processing is required for peptide activation of CLEL6 and CLEL9 based on a bioassay data using N. benthamiana leaf extracts transiently expressing GFP-CLEL6 and GFP-CLEL9 fusion proteins. This approach is, however, not quantitative and cannot exclude the possibility that structurally immature (larger or unmodified) peptide products are detected if present at high concentrations. Indeed, extracts from leaves expressing D-to-A mutants of CLEL6 and CLEL9 (that do not undergo proteolytic cleavage and posttranslational modification) still shows considerable activities (Figure 4A and B). Therefore, in addition to the bioassay experiments, mass spectroscopic analysis which can evaluate both actual structure and abundance of mature peptides is prerequisite to justify the conclusion.

2) Related to above points, there is no detailed explanation of methods for extraction of apoplastic peptides and bioassay. Explain in the Materials and methods section.

3) The authors used hypocotyl for bioassay in Figure 1, but used roots in Figure 4. This is confusing for readers. Use hypocotyl bioassay throughout the manuscript.

4) In Figure 2B, it is better to indicate theoretical molecular sizes for each possible fragments. This information greatly helps understanding of SDS-PAGE bands shown in Figure 2C, E, and F. The same is true for Figure 3B.

*Reviewer #3:*

The paper is sufficiently well written. It provides incremental data about subsequent steps in the maturation of the plant secreted signal peptides CLEL6 and 9 (also known as RGF 6 and 9, or GLV 1 and 2, respectively, in Arabidopsis) and analyzes the subcellular compartments where proteolytic events take place. The work is overall sound, except for problems explained here below (and possibly the need for replicated experiments, depending on the feedback from the authors about the growth of roots treated with "cell wall washes"). But it has a narrow scope and shows some weakness in the interpretation of the results. The manuscript is worth publishing, but I do not feel that it is a major contribution to its field.

Subsection “The cleavage for final activation occurs in a post-Golgi compartment by an aspartate-dependent SBT” fourth paragraph. Provided additional explanations are given (see below), I will trust that the experiments illustrated in Figure 4C to E show that CLEL extracts have an impact on root growth, and that there is a differential response between the wt and D>A extracts. However, the explanation of the root growth referring to the "waving phenotype" is incorrect. On dense agar plates slanted backward, wild-type untreated roots "wave", that is their normal growth pattern which is explained as a recurrent alternation between obstacle avoidance response and gravitropic response. In Whitford et al., 2012, CLEL peptides added to the medium do not induce waving, but they alter the waving pattern. In fact, the peptides induce "coiling" rather than waving: because of a reduction in gravitropism: roots often form complete loops as they grow away from the vertical.

In the experiments presented here, the control roots do not wave, but simply grow down (Figure 4C and Figure 4—figure supplement 2). The lack of waving may be due to the angle of the plate (vertical or even slanted forward), the composition of the medium, the concentration of the agar. What is reported in this manuscript is that treated roots do not follow as strictly the vertical direction compared to the controls, they meander… but they don't wave. If my interpretation is right, the text should be corrected by removing all references to "waving" and simply state that treated roots have a reduced gravitropic response.

Related to these experiments, the authors should address the following questions that are not covered in the Results or Materials and methods sections?

How were the roots grown (germination, medium, agar type and concentration, temperate(s) light and light cycles)? How exactly was the variable "changes in root growth direction" calculated? Can they mention other papers describing this method to measure gravitropism? How many independent primary roots were analyzed per treatment, in how many independent experiments and with how many independently prepared cell wall extracts? How where the apoplastic washes prepared? What does the expression "treated with equal amount of these cell wall extracts" exactly mean? There is not sufficient information in the manuscript about these experiments.

Is there not a contradiction to state that the variable pro-region of the precursor, containing the known SBT6.1 cleavage sites, is important for secretion and to show that a truncated version of the precursor NOT containing this pro-region is processed just fine (Figure 3C)? Furthermore, I understand that the authors assume [subsection “The cleavage for final activation occurs in a post-Golgi compartment by an aspartate-dependent SBT”, first paragraph] that the two parts of a precursor protein stay in the same subcellular compartment(s) after proteolytic cleavage, as the presence of the GFP signal corresponding to the protein segment on the amino terminus is interpreted as indicating where the CLEL processed peptide (on the other side of the cleavage site) also resides (Figure 3D). Is this assumption warranted?

[Editors’ note: further revisions were suggested prior to acceptance, as described below.]

Thank you for submitting your article "The biogenesis of CLEL peptides involves several processing events in consecutive compartments of the secretory pathway" for consideration by *eLife*. Your article has been reviewed by three peer reviewers, one of whom is a member of our Board of Reviewing Editors, and the evaluation has been overseen by Christian Hardtke as the Senior Editor. The following individual involved in review of your submission has agreed to reveal their identity: Yoshikatsu Matsubayashi (Reviewer #2).

The reviewers have discussed the reviews with one another and the Reviewing Editor has drafted this decision to help you prepare a revised submission.

Summary:

Although many processes in plants rely on information transmitted via small secreted proteins, peptides and/or hormones, the maturation and processing mechanisms of plant peptide hormones are poorly understood. In this manuscript, the authors show that several CLE-like peptides are processed several times, with early proteolytic cleavage within the pro-peptides variable domain is a prerequisite for continued passage through the secretory pathway and subsequent proteolytic maturation. The identification of candidate proteases for these two events was also shown.

Essential revisions:

The major issue is that the claim that SBT3.8 processes CLEL in vivo is not clearly supported by the available data. Although it is shown in this revised manuscript that SBT3.8 can cleave CLEL and that SBT3.8 is expressed, secreted and active at the right conditions, the key missing evidence is the absence of cleavage in the *sbt3.8* mutant. The data also indicate that *sbt3.8* should have a gravitropic phenotype. The authors must provide evidence for these two to be able to conclude that SBT3.8 processes CLEL in vivo. The editor expects that these experiments would be performed; if the results are that there is still some cleavage of CLELs in the mutant and/or that there is no gravitropic phenotype, this will not rule out the paper being published, but it will require that the text and summary figures are modified to accurately convey the results.

---

## [Author Response]

[Editors’ note: the authors resubmitted a revised version of the paper for consideration. What follows is the authors’ response to the first round of review.]

There was consensus that the context and rationale for the study was well laid out and that there is a need for better mechanistic understanding of how CLEL peptides are processed. The extent to which this study definitively advanced our knowledge beyond that of previous studies, however, was considered by all reviewers to be fairly modest. Certainly identification of the SBT capable of driving the next processing reaction would have made this work much more compelling.

We agree with the reviewers and the editor’s judgement that our story is incomplete as long as we cannot present the enzyme responsible for the final processing step. Using the subtilase-specific EPI inhibitor, we had shown in the original manuscript that the protease responsible for the final processing step is a subtilase. We had also shown that this processing event depends on an aspartic acid residue at the cleavage site within the precursor, and that it occurs in a post Golgi compartment. However, we had not yet identified the proposed asp-dependent protease. In this revised manuscript we show that the missing protease is subtilase SBT3.8. SBT3.8 cleaves the recombinant precursor in an asp-dependent manner. Cleavage by SBT3.8 was further shown to be pH dependent. Cleavage was only observed at acidic pH, which is consistent with the proposed localization of this cleavage event late in the secretory pathway, in a post-Golgi compartment.

Senior editor (post-decision communication)

If the authors have identified an SBT that they can show has the needed activity against a CLEL precursor and they can show, in vivo, where this enzyme is located (ideally in a stable Arabidopsis line since transient N. benthi localization has been shown to lead to artifacts), then I would be happy to reconsider this manuscript.

We show that CLEL precursors are processed at the predicted site by SBT3.8, that cleavage depends on the critical aspartate residue at the cleavage site, and that cleavage by SBT3.8 only occurs at acidic pH as it is found in late compartments of the secretory pathway including the cell wall. Using both, transient expression in *N. benthamiana* as well as stable Arabidopsis lines we show that SBT3 is targeted to the apoplast. Extracellular localization of SBT3 is fully consistent with the proposed site of precursor processing in a post-Golgi compartment, and with the acidic pH requirements of SBT3.

Reviewer #1:This is an engagingly written article about CLEL processing. The CLE-like family of peptides have been known for about 20 years; initially via genetics in Arabidopsis. There are many biological functions ascribed to this family, particularly during development-shoot meristems, root meristems, embryos all employ family members, and these proteins have also been found and studied in Maize, tomato, etc. The challenge of this family is that the active peptide are only 12 AA long and are processed and heavily modified. What I didn't appreciate until this manuscript was that, although many of the modification enzymes have been identified, the initial proteolytic processing is fairly uncharacterized. For a family as critical as this one, missing the first step is a big deal!Subtilisin proteases (SBT) are found in all forms of life. In plants, there is a large family of SBTs, but most are extracellular. This manuscript focuses on SBT6.1, one of the few TM-domain forms, and one that is also thought to have emerged before the plant/animal split. SBT6.1 was recently shown to be required for activity of CLEL6 (Ghorbani, 2016). So, this manuscript must add something new. What's new is that they show where the first SBT6.1 processing takes place and, they tracked down a second processing step that need to take place in the Golgi and they found the residue in CLEL that is needed for this second step. They then do an assay that suggests another SBT (or something that can be inhibited by EPI, anyway) is needed later on, but don't know what it is.The approach relies on several technical innovations, some of which I find very clever and others (though also clever) I am less convinced of the specificity of.e.g., I like that they set up a transient cleavage assay (N. benthi) with GFP attached to the CLEL precursors + various tags in which they could monitor process through the secretory system with live cell imaging and also by Immunoblot against GFP to see where processing occurred.I am a little worried about overexpression of a pathogen-derived EPI, in that its specificity isn't completely clear.

The EPIs were identified and characterized in the lab of Sophien Kamoun (doi:

10.1074/jbc.M400941200, Tian et al., 2005, doi:10.1186/1471-2091-6-15). EPI1 is a two domain inhibitor, consisting of one typical (EPI1b) and one atypical (EPI1a) Kazal inhibitor domain. EPI10 comprises three domains, two typical and one atypical Kazal domain. Tian et al. (doi:10.1186/1471-2091-6-15) reported that the so-called ‘atypical Kazal domain’ of EPI1 and EPI10 is highly specific for subtilases, while the ‘typical’ Kazal domains show less specificity and may also inhibit other proteases. This is why we used EPI1a (i.e. the subtilase-specific atypical domain of EPI1) in addition to EPI10 in our experiments. The same phenotype (reduced gravitropic response of the hypocotyl in EPI expressing transgenics) was observed for both inhibitors. Because of the specificity reported for EPI1a, we conclude that the phenotype is caused by the inhibition of subtilase activity. Nonetheless, we agree with the reviewer. We cannot formally exclude the possibility that EPI1a may exert some additional, unknown effects in plants. On the other hand, with the identification of SBT3.8 as the final processing protease, we confirm the predicted involvement of subtilase(s).

I do like the genetic experiments of showing that EPIs driven with CLEL6 or CLEL9 promoters leads to a phenotype similar to loss of the CLEL6/9 peptides themselves, and can be suppressed by supplying synthesized, processed CLEL.Strengths: This is well written and I am now convinced that it's important to understand the processing of the CLEL peptides and that the processing is going to be complicated. The assays are clever and the experiments appear to be done rigorously. The data are presented in a clear and straightforward manner in the figures.Weaknesses: Relies on EPIs being highly specific (and also, highly effective).

With respect to specificity, please see our response above. This is certainly true. We cannot expect to see an effect, unless the activity of endogenous subtilases is sufficiently inhibited. Therefore, expression level and binding affinity of the EPIs must be high enough to inhibit potentially redundant subtilases that are present in the respective tissue. In a previous study, we had analyzed the Kd of inhibitor binding to a plant subtilase (SBT4.13) and found it to be in the low nM range (23 nM for EPI1a, 13 nM for EPI10). For two other subtilases (SBT4.12 and SBT5.2) half-maximal inhibition (IC_50_) by EPI1a was observed at 0.8 and 6.4 nM, respectively (doi:10.1126/science.aai8550). Because of the low-nM binding/inhibition of different plant subtilases, we considered the EPIs suitable for targeting subtilase activity in vivo. The phenotype we observe for EPI1a and EPI10 over-expressors (reduced gravitropic response of the hypocotyl in the present study, a phenocopy of the abscission defect of the ida mutant in Schardon et al., 2016) suggests that concentration/affinity of the EPIs was in fact high enough to exert an effect.

Biochemical assays are in a non-native system, although I recognize that it is likely not technically feasible to do the assays in Arabidopsis and they also include bioassays in Arabidopsis, so I am less concerned about this leading to completely artifactual results.Although they demonstrated that it IS important for the peptides to be processed at the site they identified, is this a surprise based on previous work?

For SBT3.8, we now also include in vitro assays using purified recombinant enzyme (SBT3.8) and substrate (the CLEL6 precursor), as well as synthetic peptide substrates. With respect to cleavage by SBT6.1 you are absolutely right; previous work had shown that SBT6.1 is required for CLEL6 function (CLEL9 was not addressed in this study). Therefore, it was not surprising to us to see that cleavage by SBT6.1 at the predicted site is important. What is new here is not the fact that SBT6.1 is important, but why it is important. SBT6.1 does not generate the bioactive peptide; it is important because cleavage by SBT6.1 is necessary for continued passage through the secretory pathway. Therefore, all subsequent maturation events in consecutive compartments of the secretory pathway depend on the initial cleavage by SBT6.1. Other novel aspects include the characterization of the subsequent N-terminal processing event, its localization and cleavage specificity, and the identification of the protease responsible for it.

The manuscript does a nice job of demonstrating that a late Golgi SBT is likely to be involved, but then can't supply an answer as to which one it is, and I felt like there was no attempt to use other data sources (co-expression, for example) to narrow down the possibilities. All in all, I found this a bit too preliminary for an eLife paper.

We completely agree with this point – we should have addressed the identity of the SBT predicted to contribute to peptide maturation late in the secretory pathway. We followed your suggestion and used expression data to narrow down on the identity of the SBT responsible for N-terminal processing in the late Golgi. We reasoned that SBTs that are involved in the production of bioactive CLEL peptides might be upregulated in mutants that are deficient in such peptides. Therefore, we analyzed the expression of all 56 Arabidopsis SBTs in hypocotyls of the *tpst* mutant, which cannot produce sulfated peptides including CLEL6 and 9. Four SBTs were found to be upregulated, including SBT3.8. Further analysis showed that SBT3.8 is able to cleave the precursor at the predicted site in an aspartate-dependent manner (please see our response to the editor for further details on this point). With these data included, the story is much more complete and we hope that you will not consider it preliminary anymore.

The Discussion was quite long and speculative relative to the compact data section. It seems as if the authors are addressing some points of contention in the field, and I (as an outsider) don't come to the story with a particular bias that needs to be overcome. I will defer to others who are closer in the field.

We tried to be more concise in the Discussion; parts which may have been distracting to readers not directly involved in the field have been removed (about 300 words).

Reviewer #2:[…] Because the maturation mechanisms of plant peptide hormones is still poorly understood, the authors' efforts will contribute to expand our knowledge of the peptide hormone biogenesis. Indeed, their findings that cleavage within the variable domain is a prerequisite for continued passage through the secretory pathway and subsequent proteolytic maturation are interesting. However, some of the conclusions are not supported by the results, especially the assumed involvement of aspartate residue in the cleavage for final activation.1) The authors concluded that aspartate-dependent processing is required for peptide activation of CLEL6 and CLEL9 based on a bioassay data using N. benthamiana leaf extracts transiently expressing GFP-CLEL6 and GFP-CLEL9 fusion proteins. This approach is, however, not quantitative and cannot exclude the possibility that structurally immature (larger or unmodified) peptide products are detected if present at high concentrations. Indeed, extracts from leaves expressing D-to-A mutants of CLEL6 and CLEL9 (that do not undergo proteolytic cleavage and posttranslational modification) still shows considerable activities (Figure 4A and B).

We repeated this experiment using the hypocotyl bioassay (see also point 3). The effect of the D-to-A substitution was stronger in this experiment. No bioactivity was observed for the D-to-A substituted CLEL6 precursor, and that of the CLEL9 precursor was much reduced. Nevertheless, we agree with your point. There is some residual activity (at least for CLEL9) and, therefore, the aspartate is not absolutely required for bioactivity. For processing on the other hand, the aspartate is essential. SBT3.8 was found to cleave recombinant CLEL6 and a synthetic, N-terminally extended CLEL6 peptide that included 7 precursor-derived amino acids (Figure 5). Substitution of Asp by Ala abolished processing of the recombinant precursor.

As you suggested, the residual activity observed for the CLEL9 precursor in the bioassay may very well be due to immature (longer or unmodified) peptides, that retain some (probably reduced) receptor binding activity. We thus agree in that the aspartate, while required for processing by SBT3.8, is only important but not absolutely required for bioactivity. In our revisions, we paid attention not to overstate our results.

Therefore, in addition to the bioassay experiments, mass spectroscopic analysis which can evaluate both actual structure and abundance of mature peptides is prerequisite to justify the conclusion.

We tried to do this but, unfortunately, failed. We were unable to detect the fully processed and post-translationally modified CLEL6 or CLEL9 peptides in cell wall extracts by mass spectrometry. In fact, this is why we developed the bioassay as a tool for the detection of the bioactive peptides.

2) Related to above points, there is no detailed explanation of methods for extraction of apoplastic peptides and bioassay. Explain in the Materials and methods section.

We apologize for this omission. In the revised manuscript, we added some more detail in the figure legend, and the full procedure in the Materials and methods section.

3) The authors used hypocotyl for bioassay in Figure 1, but used roots in Figure 4. This is confusing for readers. Use hypocotyl bioassay throughout the manuscript.

Thank you for this suggestion. We agree that the use of two different bioassays may cause some confusion. Therefore, we repeated the experiment shown in Figure 4 using the hypocotyl bioassay. The result is fully consistent with the previous root bioassay; in fact it is even more clear. The D-to-A substitution abolished bioactivity for the CLEL6 precursor; only for the D-to-A mutant of CLEL9 precursor, some residual activity was observed (please also see our response to point 1).

4) In Figure 2B, it is better to indicate theoretical molecular sizes for each possible fragments. This information greatly helps understanding of SDS-PAGE bands shown in Figure 2C, E, and F. The same is true for Figure 3B.

Thank you for this suggestion. This point was also raised by reviewer #3, who requested to indicate more clearly what forms in Figure 2A correspond to the bands in Figure 2C, E and F. We tried different ways of how to help readers to interpret the SDS-PAGE bands and came up with the following solution. Rather than indicating the theoretical masses, we added a scheme to panel A of Figures 2 and 3, showing each of the possible fragments in a different color. Asterisks of the same color are used to mark the SDS-PAGE bands.

Reviewer #3:The paper is sufficiently well written. It provides incremental data about subsequent steps in the maturation of the plant secreted signal peptides CLEL6 and 9 (also known as RGF 6 and 9, or GLV 1 and 2, respectively, in Arabidopsis) and analyzes the subcellular compartments where proteolytic events take place. The work is overall sound, except for problems explained here below (and possibly the need for replicated experiments, depending on the feedback from the authors about the growth of roots treated with "cell wall washes"). But it has a narrow scope and shows some weakness in the interpretation of the results. The manuscript is worth publishing, but I do not feel that it is a major contribution to its field.

By adding the identification and characterization of SBT3.8 as the protease responsible for aspartate- and pH-dependent processing at the N–terminus of CLEL peptides we extended the scope considerably. Considering the limited knowledge we have about the proteases and factors that govern proteolytic maturation of peptide hormones, we feel that our study now makes an important contribution to the field.

Subsection “The cleavage for final activation occurs in a post-Golgi compartment by an aspartate-dependent SBT” fourth paragraph. Provided additional explanations are given (see below), I will trust that the experiments illustrated in Figure 4C to E show that CLEL extracts have an impact on root growth, and that there is a differential response between the wt and D>A extracts. However, the explanation of the root growth referring to the "waving phenotype" is incorrect. On dense agar plates slanted backward, wild-type untreated roots "wave", that is their normal growth pattern which is explained as a recurrent alternation between obstacle avoidance response and gravitropic response. In Whitford et al., 2012, CLEL peptides added to the medium do not induce waving, but they alter the waving pattern. In fact, the peptides induce "coiling" rather than waving: because of a reduction in gravitropism: roots often form complete loops as they grow away from the vertical.In the experiments presented here, the control roots do not wave, but simply grow down (Figure 4C and Figure 4—figure supplement 2). The lack of waving may be due to the angle of the plate (vertical or even slanted forward), the composition of the medium, the concentration of the agar. What is reported in this manuscript is that treated roots do not follow as strictly the vertical direction compared to the controls, they meander… but they don't wave. If my interpretation is right, the text should be corrected by removing all references to "waving" and simply state that treated roots have a reduced gravitropic response.

We fully agree with this interpretation. Since ‘waving’ is the normal behavior of wild type roots, we cannot refer to the difference between roots treated with wt and D>A extracts as the ‘waving phenotype’. The suggested wording that ‘treated roots show a reduced gravitropic response’ is much more appropriate. Thank you for this suggestion. However, on request of reviewer #2, we replaced the root growth bioassay by a hypocotyl bioassay in the revised manuscript. Therefore, the changes to the subsection “The cleavage for final activation occurs in a post-Golgi compartment by aspartate dependent subtilase SBT3.8” and Figure 4—figure supplement 2 legend, are now obsolete.

Related to these experiments, the authors should address the following questions that are not covered in the Results or Materials and methods sections?How were the roots grown (germination, medium, agar type and concentration, temperate(s) light and light cycles)? How exactly was the variable "changes in root growth direction" calculated? Can they mention other papers describing this method to measure gravitropism? How many independent primary roots were analyzed per treatment, in how many independent experiments and with how many independently prepared cell wall extracts?

Some of the points raised here (those directly related to the root growth bioassay) are obsolete (see point above). The hypocotyl bioassay was performed in three independent experiments using independent cell wall extracts. The data shown in Figure 4 are the representative result of one of these experiments. This information was included in the figure legend.

How where the apoplastic washes prepared? What does the expression "treated with equal amount of these cell wall extracts" exactly mean?

The CLEL precursors and D-to-A mutants were expressed as GFP fusion proteins. Therefore, we used GFP fluorescence to quantify expression levels. ‘equal amounts of cell wall extracts’ referred to equal amounts of GFP fluorescence as a measure of protein expression. We agree that this statement is unclear; it was rephrased in the revised manuscript (subsection “The cleavage for final activation occurs in a post-Golgi compartment by aspartate-dependent subtilase SBT3.8”, fourth paragraph).

There is not sufficient information in the manuscript about these experiments.

We apologize for this omission. In the revised manuscript, we added some more detail in the figure legend, and the full procedure in the Materials and methods section.

Is there not a contradiction to state that the variable pro-region of the precursor, containing the known SBT6.1 cleavage sites, is important for secretion and to show that a truncated version of the precursor NOT containing this pro-region is processed just fine (Figure 3C)?

This is a good observation. The fact that the truncated precursor lacking most of the prodomain is efficiently processed (Figure 3C) does indeed suggest that it is a minor fraction that is retained intracellularly (Figure 3D). The effect caused by mutation of the SBT6.1 cleavage sites is clearly more pronounced, as both secretion and processing are affected. We therefore toned down the relevance of the prodomain as compared to cleavage by SBT6.1 for secretion.

Furthermore, I understand that the authors assume [subsection “The cleavage for final activation occurs in a post-Golgi compartment by an aspartate-dependent SBT”, first paragraph] that the two parts of a precursor protein stay in the same subcellular compartment(s) after proteolytic cleavage, as the presence of the GFP signal corresponding to the protein segment on the amino terminus is interpreted as indicating where the CLEL processed peptide (on the other side of the cleavage site) also resides (Figure 3D). Is this assumption warranted?

In this example (the Δ-Sec construct), most of the GFP signal is extracellular (Figure 3D), and most of the precursor is processed (Figure 3C). Therefore, in this particular case, the two parts of the precursor are expected to be in the same compartment (the cell wall). However, we do not assume that this is generally the case. For the KDEL or XylT-tagged precursors, for example, GFP fluorescence indicates that the N-terminal part of the protein is retained in the ER or Golgi, respectively. Cleavage of these precursors would separate the C-terminus from the retention signals and, therefore, any cleavage product would be expected to continue along the secretory pathway, potentially ending up in the cell wall.

[Editors’ note: what follows is the authors’ response to the second round of review.]

Essential revisions:The major issue is that the claim that SBT3.8 processes CLEL in vivo is not clearly supported by the available data. Although it is shown in this revised manuscript that SBT3.8 can cleave CLEL and that SBT3.8 is expressed, secreted and active at the right conditions, the key missing evidence is the absence of cleavage in the sbt3.8 mutant. The data also indicate that sbt3.8 should have a gravitropic phenotype. The authors must provide evidence for these two to be able to conclude that SBT3.8 processes CLEL in vivo. The editor expects that these experiments would be performed; if the results are that there is still some cleavage of CLELs in the mutant and/or that there is no gravitropic phenotype, this will not rule out the paper being published, but it will require that the text and summary figures are modified to accurately convey the results.

We compared CLEL6 precursor processing in wild type and the *sbt3.*8 loss-of-function mutant. Since we have no way of analyzing the processing of endogenous peptide precursors in planta, we prepared cell wall exudates of wild type plants and *sbt3.8* mutants. Wild-type exudates efficiently cleaved the CLEL6 precursor, *sbt3.8* mutant exudates did not. CLEL6 precursor cleavage thus depends on SBT3.8 under these conditions. The data were added as Figure 5—figure supplement 2B. In order not to overstate our results, we write in the text:’ these data support a role for CLEL6 maturation in vivo’ (subsection “The cleavage for final activation occurs in a post-Golgi compartment by aspartate dependent subtilase SBT3.8”, last paragraph). We do not claim that SBT3.8 is the only protease capable of CLEL6 precursor processing. In fact, evidence shows that there must be redundant activities: the gravitropic response of the hypocotyl turned out to be unaffected in the *sbt3.8* mutant (Figure 5—figure supplement 2C). Therefore, to acknowledge the presence of redundant activities, we changed the wording in the Abstract and Discussion. Also, we revised the model in Figure 6, including ‘other D-dependent subtilases’ in addition to SBT3.8, for the final processing event.